# AutoBalance: Optimized Loss Functions for Imbalanced Data

**Mingchen Li     Xuechen Zhang**
University of California, Riverside
{mli176,xzhan394}@ucr.edu

**Christos Thrampoulidis**
University of British Columbia
cthrampo@ece.ubc.edu.ca

**Jiasi Chen**
University of California, Riverside
jiasi@cs.ucr.edu

**Samet Oymak**
University of California, Riverside
oymak@ece.ucr.edu

## Abstract

Imbalanced datasets are commonplace in modern machine learning problems. The presence of under-represented classes or groups with sensitive attributes results in concerns about generalization and fairness. Such concerns are further exacerbated by the fact that large capacity deep nets can perfectly fit the training data and appear to achieve perfect accuracy and fairness during training, but perform poorly during test. To address these challenges, we propose AutoBalance, a bi-level optimization framework that automatically designs a training loss function to optimize a blend of accuracy and fairness-seeking objectives. Specifically, a lower-level problem trains the model weights, and an upper-level problem tunes the loss function by monitoring and optimizing the desired objective over the validation data. Our loss design enables personalized treatment for classes/groups by employing a parametric cross-entropy loss and individualized data augmentation schemes. We evaluate the benefits and performance of our approach for the application scenarios of imbalanced and group-sensitive classification. Extensive empirical evaluations demonstrate the benefits of AutoBalance over state-of-the-art approaches. Our experimental findings are complemented with theoretical insights on loss function design and the benefits of train-validation split. All code is available open-source.

## 1 Introduction

Recently, deep learning, large datasets, and the evolution of computing power have led to unprecedented success in computer vision, and natural language processing [15, 39, 62]. This success is partially driven by the availability of high-quality datasets, built by carefully collecting a sufficient number of samples for each class. In practice, real-world datasets are frequently imbalanced and exhibit long-tailed behavior, necessitating a careful treatment of the minorities [20, 53, 23]. Indeed, modern classification tasks can involve thousands of classes, so it is perhaps intuitive that some classes should be over/under-represented compared to others. Besides class imbalance, minorities can also appear at the feature-level; for instance, the specific values of the features of an example can vary depending on that example's membership in certain sensitive or protected groups, e.g. race, gender, disabilities (see also Figure 1a). In scenarios where imbalances are induced by heterogeneous client datasets (e.g., in the context of federated learning), addressing these imbalances can help ensure that a machine learning model works well for all clients, rather than just those that generate the majority of the training data. This rich set of applications motivate the careful treatment of imbalanced datasets.

In the imbalanced classification literature, the recurring theme is maximizing a fairness-seeking objective, such as balanced accuracy. Unlike standard accuracy, which can be dominated by the majorities, a fairness-seeking objective seeks to promote examples from minorities, and downweigh

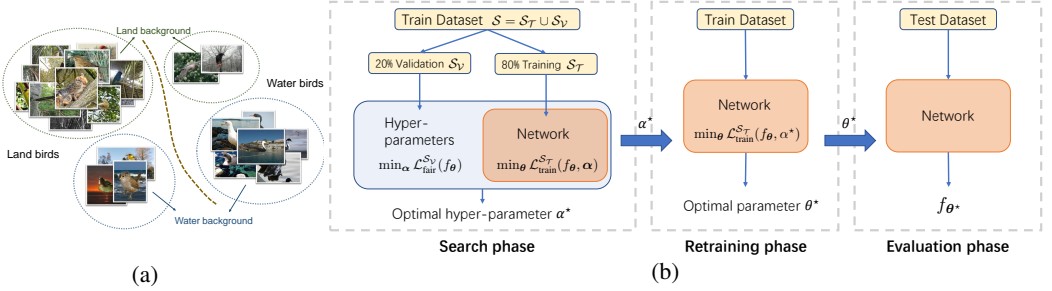

Figure 1: (a) Example group-imbalance on the Waterbirds dataset [63, 73]. Groups correspond to the distinct background types, while classes are distinct bird types. (b) Framework overview. The search phase conducts a bilevel optimization to design the optimal training loss function parameterized by $\boldsymbol{\alpha}^\star$ by minimizing the validation loss, using a train-validation split (e.g. 80%-20%). The retrain phase uses the original training data and $\boldsymbol{\alpha}^\star$ to obtain the optimal model parameters $\boldsymbol{\theta}^\star$. The evaluation phase predicts the test data using $\boldsymbol{\theta}^\star$.

examples from majorities. Here, note that there is a distinction between the test and training objectives. While the overall goal is typically to maximize a non-differentiable objective such as balanced accuracy on the test set, during training, we use a differentiable proxy for this, such as weighted cross-entropy. Thus, the fundamental question of interest is:

> How to design a training loss to maximize a fairness-seeking objective on the test set?

A classical answer to this question is to use a Bayes-consistent loss functions. For instance, weighted cross-entropy (e.g., each class gets a different weight, see Sec. 2) is traditionally a good choice for optimizing weighted accuracy objectives. Unfortunately, this intuition starts to break down when the training problem is overparameterized, which is a common practice in deep learning: in essence, for large capacity deep nets, the training process can perfectly fit to the data, and training loss is no longer indicative of test error. In fact, recent works [8, 38] show that weighted cross-entropy has minimal benefit to balanced accuracy, and instead alternative methods based on margin adjustment can be effective (namely, by ensuring that minority classes are further away from decision boundary). These ideas led to the development of a parametric cross-entropy function $\ell(y, f(\boldsymbol{x})) = w_y \log \big(1 + \sum_{k \neq y} e^{l_k - l_y} \cdot e^{\Delta_k f_k(\boldsymbol{x}) - \Delta_y f_y(\boldsymbol{x})}\big)$, which allows for a personalized treatment of the individual classes via the design parameters $(w_k, l_k, \Delta_k)_{k=1}^K$ [8, 38, 53, 35, 67]. Here, $w_k$ is the classical weighting term whereas $l_k$ and $\Delta_k$ are additive and multiplicative logit adjustments. However, despite these developments, it is unclear how such parametric cross-entropy functions can be tuned for use for different fairness objectives, for example to tackle class or group imbalances. The works by [8, 53] provide theoretically-motivated choices for $(w_k, l_k)$, while [38] argues that $(w_k, l_k)$ is not as effective as $\Delta_k$ in the interpolating regime of zero training error and proposes the simultaneous use of all three different parameter types. However, these works do not provide an optimized loss function that can be systematically tailored for different fairness objectives, such as balanced accuracy common in class imbalanced scenarios, or equal opportunity [23, 17] which is relevant in group-sensitive settings.

In this work, we address these shortcomings by designing the loss function within the optimization *in a principled fashion*, to handle different fairness-seeking objectives. Our main idea is to use bi-level optimization, where the model weights are optimized over the training data, and the loss function is automatically tuned by monitoring the validation loss. Our core intuition is that unlike training data, the validation data is difficult to fit and will provide a consistent estimator of the test objective.

**Contributions.** Based on this high-level idea, this paper takes a step towards a systematic treatment of imbalanced learning problems with contributions along several fronts: state-of-the-art performance, data augmentation, applications to different imbalance types, and theoretical intuitions. Specifically:
• We introduce *AutoBalance* —a bilevel optimization framework— that designs a fairness-seeking loss function by jointly training the model and the loss function hyperparameters in a systematic way (Figure 1b, Section 2). We introduce novel strategies that narrow down the search space to improve convergence and avoid overfitting. To further improve the performance, our design also incorporates *data augmentation policies* personalized to subpopulations (classes or groups). We demonstrate the benefits of AutoBalance when optimizing various fairness-seeking objectives over the state-of-the-art, such as logit-adjustment (LA) [53] and label-distribution-aware margin (LDAM) [8] losses. The code is available online [42].

**Algorithm 1:** AutoBalance via Bilevel Optimization

---

**Input:** Model $f_{\boldsymbol{\theta}}$ with weights $\boldsymbol{\theta}$, dataset $\mathcal{S} = \mathcal{S}_{\mathcal{T}} \cup \mathcal{S}_{\mathcal{V}}$, step sizes $\eta_{\boldsymbol{\alpha}}$ & $\eta_{\boldsymbol{\theta}}$, # iterations $t_2 > t_1$

1  Initialize $\boldsymbol{\alpha}$ with $\ell_{\text{fair}}(\cdot) = \ell_{\text{train}}(\cdot; \boldsymbol{\alpha})$      `// Consistent initialization`

2  Train $\boldsymbol{\theta}$ for $t_1$ iterations ($\boldsymbol{\alpha}$ is fixed)      `// ``Search Phase:`` Starts with warm-up`

3  **for** $i \leftarrow t_1$ **to** $t_2$ **do**

4     |  Sample training batch $\mathcal{B}_{\mathcal{T}}$ from $\mathcal{S}_{\mathcal{T}}$;

5     |  $\mathcal{B}_{\mathcal{T}} \leftarrow \mathcal{A}(\mathcal{B}_{\mathcal{T}})$      `// Apply class-personalized augmentation`

6     |  $\boldsymbol{\theta} \leftarrow \boldsymbol{\theta} - \eta_{\boldsymbol{\theta}} \nabla_{\boldsymbol{\theta}} \mathcal{L}_{\text{train}}^{\mathcal{B}_{\mathcal{T}}}(f_{\boldsymbol{\theta}}; \boldsymbol{\alpha})$

7     |  Sample validation batch $\mathcal{B}_{\mathcal{V}}$ from $\mathcal{S}_{\mathcal{V}}$;

8     |  Compute hyper-gradient $\nabla_{\boldsymbol{\alpha}} \mathcal{L}_{\text{fair}}^{\mathcal{B}_{\mathcal{V}}}(f_{\boldsymbol{\theta}})$      `// via Approx. Implicit Differentiation`

9     |  $\boldsymbol{\alpha} \leftarrow \boldsymbol{\alpha} - \eta_{\boldsymbol{\alpha}} \nabla \mathcal{L}_{\text{fair}}^{\mathcal{B}_{\mathcal{V}}}(f_{\boldsymbol{\theta}})$      `// Update loss function hyper-parameters`

10  **end**

11  Set $\boldsymbol{\alpha}_{\star} \leftarrow \boldsymbol{\alpha}$, $\mathcal{S}_{\mathcal{T}} \leftarrow \mathcal{S}$, reset weights $\boldsymbol{\theta}$      `// ``Retraining Phase:`` Use all data and `$\boldsymbol{\alpha}_{\star}$

12  Train $\boldsymbol{\theta}$ for $t_2$ iterations using $\boldsymbol{\alpha}_{\star}$

**Result:** The final model $\boldsymbol{\theta}_{\star} \leftarrow \boldsymbol{\theta}$ and hyper-parameters $\boldsymbol{\alpha}_{\star}$

---

● Extensive experiments provide several takeaways (Section 3). First, AutoBalance discovers loss functions from scratch that are consistent with theory and intuition: hyperparameters of the minority classes evolve to upweight the training loss of minority to promote them. Second, the impact of individual design parameters in the loss function is revealed, with the additive adjustment $l_k$ and multiplicative adjustment $\Delta_k$ synergistically improving the fairness objective. Third, personalized data augmentation can further improve the performance over a single generic augmentation policy.

● Beyond class imbalance, we consider applications of loss function design to the group-sensitive setting (Section 4). Our experiments show that AutoBalance consistently outperforms various baselines, leading to a more efficient Pareto-frontier of accuracy-fairness tradeoffs.

### 1.1   Problem Setup for Class Imbalance

We first focus on the label-imbalance problem. The extension to the group-imbalanced setting (approach, algorithms, evaluations) is deferred to Section 4. Let $[K]$ denote the set $\{1, \dots, K\}$. Suppose we have a dataset $\mathcal{S} = (\boldsymbol{x}_i, y_i)_{i=1}^n$ sampled i.i.d. from a distribution $\mathcal{D}$ with input space $\mathcal{X}$ and $K$ classes. For a training example $(\boldsymbol{x}, y)$, $\boldsymbol{x} \in \mathcal{X}$ is the input feature and $y \in [K]$ is the output label. Let $f : \mathcal{X} \to \mathbb{R}^K$ be a model that outputs a distribution over classes and let $\hat{y}_f(\boldsymbol{x}) = \arg\max_{i \in [K]} f(\boldsymbol{x})$. The standard classification error is denoted by $\mathcal{E}(f) = \mathbb{P}_{\mathcal{D}}[y \neq \hat{y}_f(\boldsymbol{x})]$. For a loss function $\ell(y, \hat{y})$ (e.g. cross-entropy), we similarly denote

$$\text{Population risk: } \mathcal{L}(f) = \mathbb{E}_{\mathcal{D}}[\ell(y, \hat{y}_f(\boldsymbol{x}))] \quad \text{and} \quad \text{Empirical risk: } \mathcal{L}^{\mathcal{S}}(f) = \frac{1}{n} \sum_{i=1}^n \ell(y_i, \hat{y}_f(\boldsymbol{x}_i)).$$

✓ **Setting: Imbalanced classes.** Define the frequency of the $k$'th class via $\boldsymbol{\pi}_k = \mathbb{P}_{(\boldsymbol{x},y) \sim \mathcal{D}}(y = k)$. Label/class-imbalance occurs when the class frequencies differ substantially, i.e., $\max_{i \in [K]} \boldsymbol{\pi}_i \gg \min_{i \in [K]} \boldsymbol{\pi}_i$. Let us introduce

$$\text{Balanced risk: } \mathcal{L}_{\text{bal}}(f) = \frac{1}{K} \sum_{k=1}^K \mathcal{L}_k(f) \quad \text{and} \quad \text{Class-conditional risk: } \mathcal{L}_k(f) = \mathbb{E}_{\mathcal{D}_k}[\ell(y, \hat{y}_f(\boldsymbol{x}))].$$

Similarly, let $\mathcal{E}_k(f)$ be the class-conditional classification error and $\mathcal{E}_{\text{bal}}(f) := (1/K) \sum_{k=1}^K \mathcal{E}_k(f)$ be the *balanced error*. In this setting, rather than the standard test error $\mathcal{E}(f)$, our goal is to the minimize balanced error. At a high-level, we propose to do this by designing an imbalance-aware training loss that maximizes balanced validation accuracy.

## 2   Methods: Loss Functions, Search Space Design, and Bilevel Optimization

Our main goal in this paper is automatically designing loss functions to optimize target objectives for imbalanced learning (e.g., Settings A and B). We will employ a parametrizable family of loss functions that can be tailored to the needs of different classes or groups. Cross-entropy variations have been proposed by [44, 35, 16] to optimize balanced objectives. Our design space will utilize recent works which introduce Label-distribution-aware margin (LDAM) [8], Logit-adjustment (LA) [53], Class-dependent temperatures (CDT) [67], Vector scaling (VS) [38] losses. Specifically, we

build on the following parametric loss function controlled by three vectors $\boldsymbol{w}, \boldsymbol{l}, \boldsymbol{\Delta} \in \mathbb{R}^K$:

$$\ell(y, f(\boldsymbol{x})) = w_y \log \big(1 + \sum_{k \neq y} e^{l_k - l_y} \cdot e^{\Delta_k f_k(\boldsymbol{x}) - \Delta_y f_y(\boldsymbol{x})}\big). \tag{2.1}$$

Here, $w_y$ enables conventional weighted CE and $l_y$, and $\Delta_y$ are additive and multiplicative adjustments to the logits. This choice is same as the VS-loss introduced in [38], which borrows the $\boldsymbol{\Delta}$ term from [67] and $\boldsymbol{l}$ term from [8, 53]. [53] makes the observation that we can use $\boldsymbol{l}$ rather than $\boldsymbol{w}$ while ensuring Fisher consistency in balanced error. We make the following complementary observation.

**Lemma 1** *Parametric loss function* (2.1) *is not consistent for standard or balanced errors if there are distinct multiplicative adjustments i.e.* $\Delta_i \neq \Delta_j$ *for some* $i, j \in [K]$.

While consistency is a desirable property, it is intuitively more critical during the earlier phase of the training where the training risk is more indicative of the test risk. In the interpolating regime of zero-training error, [38] shows that $\boldsymbol{w}, \boldsymbol{l}$ can be ineffective and multiplicative $\boldsymbol{\Delta}$-adjustment can be more favorable. Our algorithm will be initialized with a consistent weighted-CE; however, we will allow the algorithm to automatically adapt to the interpolating regime by tuning $\boldsymbol{l}$ and $\boldsymbol{\Delta}$.

**Proposed training loss function.** For our algorithm, we will augment (2.1) with *data augmentation* that can be *personalized to distinct classes*. Let us denote the data augmentation policies by $\mathcal{A} = (\mathcal{A}_y)_{y=1}^K$ where each $\mathcal{A}_y$ stochastically augments an input example with label $y$. Additionally, we clamp $\boldsymbol{\Delta}_i$ with the sigmoid function $\sigma$ to limit its range to (0,1) to ensure non-negativity. To this end, our loss function for the lower-level optimization (over training data) is as follows:

$$\ell_{\text{train}}(y, \boldsymbol{x}, f; \boldsymbol{\alpha}) = -\mathbb{E}_{\mathcal{A}} \left[ w_y \log \left( \frac{e^{\sigma(\Delta_y) f_y(\mathcal{A}_y(\boldsymbol{x})) + l_y}}{\sum_{i \in [K]} e^{\sigma(\Delta_i) f_i(\mathcal{A}_y(\boldsymbol{x})) + l_i}} \right) \right]. \tag{2.2}$$

Here, $\boldsymbol{\alpha}$ is the set of hyperparameters of the loss function that we wish to optimize, specifically $\boldsymbol{\alpha} = [\boldsymbol{w}, \boldsymbol{l}, \boldsymbol{\Delta}, \text{param}(\mathcal{A})]$. $\text{param}(\mathcal{A})$ is the parameterization of the augmentation policies $(\mathcal{A}_y)_{y \in [K]}$.

*Personalized data augmentation (PDA).* Remarkable benefits of data augmentation techniques provide a natural motivation to investigate whether one can benefit from learning class-personalized augmentation policies. The PDA idea relates to SMOTE [9], where the minority class is over-sampled by creating synthetic examples. To formalize the benefits of PDA, consider a spherical augmentation strategy where $\mathcal{A}_y(\boldsymbol{x})$ samples a vector uniformly from an $\ell_2$-ball of radius $\varepsilon_y$ around $\boldsymbol{x}$. As visualized in Figure 2 for a linear classifier, if the augmentation strengths of both classes are equal, the max-margin classifier is not affected by the application of the data augmentation and remains identical. Thus, augmentation has no benefit. However by applying a

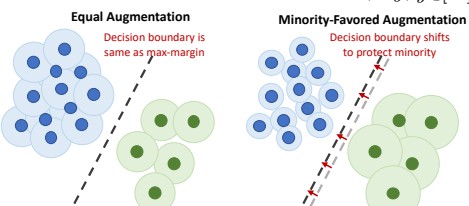

Figure 2: Data augmentation can shift the decision boundary to benefit the minority class by providing a larger margin. Lemma 2 establishes an equivalence between spherical data augmentation and parametric cross-entropy loss.

stronger augmentation on minority, the decision boundary is shifted to protect minority which can provably benefit the balanced accuracy [38]. The following intuitive observation links the PDA to parametric loss (2.1).

**Lemma 2** *Consider a binary classification task with labels* 0 *and* 1 *and a linearly separable training dataset. For any parametric loss* (2.1) *choices of* $(l_i, \Delta_i, w_i)_{i=0}^1$, *there exists spherical augmentation strengths for minority/majority classes so that, without regularization,* **optimizing the logistic loss with personalized augmentations returns the same classifier as optimizing** (2.1).

This lemma is similar in flavor to [31], which considers a larger uncertainty set around the minority class. But, as discussed in the appendix, Lemma 2 is relevant in the overparameterized regime whereas the approach of [31] is ineffective for separable data [51]. Algorithmically, the augmentations that we consider are much more flexible than the $\ell_p$-balls of [31] and our experiments showcase the value of our approach in state-of-the-art multiclass settings. Besides, note that the (theoretical) benefits of PDA can go well-beyond Lemma 2 by leveraging the invariances [10, 14] (via rotation, translation).

## 2.1 Proposed Bilevel Optimization Method

We formulate the loss function design as a bilevel optimization over hyperparameters $\boldsymbol{\alpha}$ and a hypothesis set $\mathcal{F}$. We split the dataset $\mathcal{S}$ into training $\mathcal{S}_{\mathcal{T}}$ and validation $\mathcal{S}_{\mathcal{V}}$ sets with $n_{\mathcal{T}}$ and $n_{\mathcal{V}}$

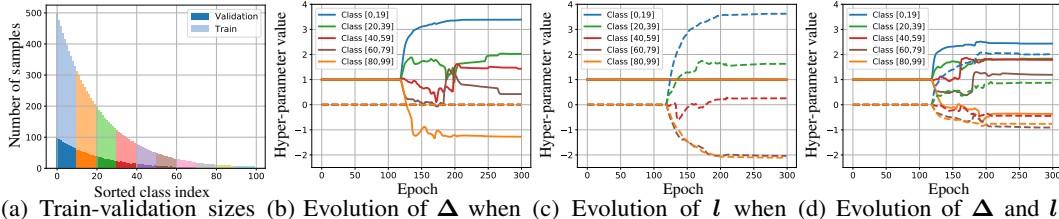

(a) Train-validation sizes for CIFAR100-LT clusters    (b) Evolution of $\boldsymbol{\Delta}$ when only training $\boldsymbol{\Delta}$    (c) Evolution of $\boldsymbol{l}$ when only training $\boldsymbol{l}$    (d) Evolution of $\boldsymbol{\Delta}$ and $\boldsymbol{l}$ when optimizing jointly

Figure 3: (a) Visualizing class clustering and train-validation split. (b), (c), (d) Evolution of loss function parameters $\boldsymbol{l}$, $\boldsymbol{\Delta}$ over epochs for CIFAR100-LT where solid curves and dashed curves corresponds to $\boldsymbol{\Delta}$ and $\boldsymbol{l}$ respectively. We display average value of 20 classes for better visualization. Based on theory, the minority classes should be assigned a larger margin. During the initial 120 epochs, we use weighted cross-entropy training and AutoBalance kicks in after epoch 120. Observe that, AutoBalance does indeed learn larger parameters $(l_y, \Delta_y)$ for minority class clusters (each containing 20 classes) consistent with theoretical intuition. In all Figures (b), (c), (d), by the end of training, the colors are ordered according to the class frequency. However, when $\Delta_y$ is trained jointly with $l_y$ (Fig d), the training is more stable compared to training $\Delta_y$ alone (Fig b). Thus, besides its accuracy benefits in Table 1, $l_y$ also seems to have optimization benefits.

examples respectively. Let $\mathcal{E}_{\text{fair}}$ be the desired test-error objective. When $\mathcal{E}_{\text{fair}}$ is not differentiable, we use a weighted cross-entropy (CE) loss function $\ell_{\text{fair}}(y, \hat{y})$ chosen to be consistent with $\mathcal{E}_{\text{fair}}$. For instance, $\mathcal{E}_{\text{fair}}$ could be a superposition of standard and balanced classification errors, i.e. $\mathcal{E}_{\text{fair}} = (1-\lambda)\mathcal{E} + \lambda\mathcal{E}_{\text{bal}}$. Then, we simply choose $\ell_{\text{fair}} = (1-\lambda)\text{CE} + \text{CE}_{\text{bal}}$. The hyperparameter $\boldsymbol{\alpha}$ aims to minimize the loss $\ell_{\text{fair}}$ over the validation set $\mathcal{S}_{\mathcal{V}}$ and the hypothesis $f \in \mathcal{F}$ aims to minimize the training loss (2.2) as follows:

$$\min_{\boldsymbol{\alpha}} \mathcal{L}_{\text{fair}}^{\mathcal{S}_{\mathcal{V}}}(f_{\boldsymbol{\alpha}}) \quad \text{WHERE} \quad f_{\boldsymbol{\alpha}} = \arg\min_{f \in \mathcal{F}} \mathcal{L}_{\text{train}}^{\mathcal{S}_{\mathcal{T}}}(f; \boldsymbol{\alpha}) := \frac{1}{n_{\mathcal{T}}} \sum_{i=1}^{n_{\mathcal{T}}} \ell_{\text{train}}(y_i, \boldsymbol{x}_i, f; \boldsymbol{\alpha}). \quad (2.3)$$

Here, $\mathcal{L}_{\text{fair}}/\mathcal{L}_{\text{fair}}^{\mathcal{S}_{\mathcal{V}}}$ are the test/validation risks associated with $\ell_{\text{fair}}$, e.g. $\mathcal{L}_{\text{fair}} = \mathbb{E}_{\mathcal{D}}[\ell_{\text{fair}}]$ as in Section 1.1. Algorithm 1 summarizes our approach and highlights the key components. The training loss $\ell_{\text{train}}(\cdot; \boldsymbol{\alpha})$ is also initialized to be consistent with $\mathcal{E}_{\text{fair}}$ (e.g., same as $\ell_{\text{fair}}$). In line with the literature on bilevel optimization, we will refer to the two minimizations of the validation and training losses in (2.3) as upper and lower level optimizations, respectively.

**Implicit Differentiation and Warm-up training.** For a loss function parameter $\boldsymbol{\alpha}$, the hyper-gradient can be written via the chain-rule $\frac{\partial \mathcal{L}_{\text{fair}}(\boldsymbol{\theta}^\star)}{\partial \boldsymbol{\alpha}} = \frac{\partial \mathcal{L}_{\text{fair}}}{\partial \boldsymbol{\alpha}} + \frac{\partial \mathcal{L}_{\text{fair}}}{\partial \boldsymbol{\theta}^\star} \frac{\partial \boldsymbol{\theta}^\star}{\partial \boldsymbol{\alpha}}$ [46]. Here, $\boldsymbol{\theta}^\star$ is the solution of the lower-level problem. We note that $\partial \mathcal{L}_{\text{fair}}/\partial \boldsymbol{\alpha} = 0$ since $\boldsymbol{\alpha}$ does not appear within the upper-level loss. Also observe that $\partial \mathcal{L}_{\text{fair}}(\boldsymbol{\theta}^\star)/\partial \boldsymbol{\theta}^\star$ can be directly computed by taking the gradient. To compute $\partial \boldsymbol{\theta}^\star/\partial \boldsymbol{\alpha}$, we follow the recent work [46] and employ the Implicit Function Theorem (IFT). If there exists a fixed point $(\boldsymbol{\theta}^\star, \boldsymbol{\alpha}^\star)$ that satisfies $\partial \mathcal{L}_{\text{train}}(\boldsymbol{\theta}^\star, \boldsymbol{\alpha}^\star)/\partial \boldsymbol{\theta} = 0$ and regularity conditions are satisfied, then around $\boldsymbol{\alpha}^\star$, there exists a function $\boldsymbol{\theta}(\boldsymbol{\alpha})$ such that $\boldsymbol{\theta}(\boldsymbol{\alpha}^\star) = \boldsymbol{\theta}^\star$ and we also have $\frac{\partial \boldsymbol{\theta}}{\partial \boldsymbol{\alpha}} = (\frac{\partial^2 \mathcal{L}_{\text{train}}}{\partial \boldsymbol{\theta}^2})^{-1} \frac{\partial^2 \mathcal{L}_{\text{train}}}{\partial \boldsymbol{\theta} \partial \boldsymbol{\alpha}}$. However, directly computing inverse Hessian $(\frac{\partial^2 \mathcal{L}_{\text{train}}}{\partial \boldsymbol{\theta}^2})^{-1}$ is usually time consuming or even impossible for modern neural networks which have millions of parameters. To compute the hyper-gradient while avoiding extensive computation, we approximate the inverse Hessian via the Neumann series, which is widely used for inverse Hessian estimation [43, 46]. Finally, the warm-up phase of our method (Line 2 of Algo. 1) is essential to guarantee that the IFT assumption $\frac{\partial \mathcal{L}_{\text{train}}(\boldsymbol{\theta}^\star, \boldsymbol{\alpha}^\star)}{\partial \boldsymbol{\theta}} = 0$ is approximately satisfied.

**Why Bilevel Optimization?** We choose differentiable optimization over alternative hyperparameter tuning methods because our hyperparameter space is continuous and potentially large (e.g. in the order of $K = 8,142$ for iNaturalist). In our experiments, the runtime of our method was typically 4∼5 times that of standard training (with known hyperparameters). Intuitively, the runtime is at least twice due to our use of separate search and retraining phases. In the appendix, we also compare against alternative approaches (specifically SMAC of [26]) and found that our approach is faster and more accurate on CIFAR10-LT.

### 2.2 Reducing the Hyperparameter Search Space and the Benefits of Validation Set

Suppose we wish to optimize the hyperparameter $\boldsymbol{\alpha} = (w_y, l_y, \Delta_y)_{y=1}^K$ of the parametric loss (2.1). An important challenge is the dimensionality of $\boldsymbol{\alpha}$, which is proportional to the number of classes $K$, as we need a triplet $(w_y, l_y, \Delta_y)$ for each class. For instance, ImageNet has $K = 1,000$

whereas iNaturalist has $K = 8,142$ classes resulting in high-dimensional hyperparameters. In our experiments, we found that directly optimizing over such large spaces leads to convergence issues likely because of the difficulty of hypergradient estimation. Additionally, with large number of hyperparameters there is increased concern for validation overfitting. This is especially so for the tail classes (e.g. the smallest class in CIFAR100-LT has only 1 validation example with an 80-20% split). On the other hand, it is well-known in AutoML literature (e.g. neural architecture search [45], AutoAugment [12]) that designing a good search space is critical for attaining faster convergence and good validation accuracy. To this end, we propose *subspace-based search spaces* for hyperparameters $(w_y, l_y, \Delta_y)$. To explain the idea, consider the logit-adjustment parameters $\boldsymbol{l} = [l_1 \ \ldots \ l_K]$ and $\boldsymbol{\Delta} = [\Delta_1 \ \ldots \ \Delta_K]$. We propose representing these $K$ dimensional vectors via $K' < K$ dimensional embeddings $\boldsymbol{l}', \boldsymbol{\Delta}'$ as follows

$$\boldsymbol{\Delta} = \boldsymbol{D_\pi}\boldsymbol{\Delta}' \quad \text{and} \quad \boldsymbol{l} = \boldsymbol{D_\pi}\boldsymbol{l}' \qquad \text{where} \qquad \boldsymbol{l}', \boldsymbol{\Delta}' \in \mathbb{R}^{K'}.$$

Here, $\boldsymbol{D_\pi} \in \mathbb{R}^{K \times K'}$ is a *frequency-aware dictionary* matrix that we design, and the range space of $\boldsymbol{D_\pi}$ becomes the hyperparameter search space. In our algorithm, we cluster the classes in terms of their frequency and assign the same hyperparameter to classes with similar frequencies. To be concrete, if each cluster has size $C$, then $K' = \lceil K/C \rceil$. For CIFAR10-LT, CIFAR100-LT, ImageNet-LT and iNaturalist, we use $C = \{1, 10, 20, 40\}$ respectively. In this scheme, each column $\boldsymbol{d}$ of the matrix $\boldsymbol{D_\pi}$ is the indicator function of one of the $K'$ clusters, i.e. $\boldsymbol{d}_i = 1$ if $i$th class is within the cluster and $0$ otherwise. Clusters are pictorially illustrated in Fig. 3a. Finally, we remark that the specific hyperparameter choices of CDT [67], LA [53], and VS [38] losses in the corresponding papers, can be viewed as specific instances of the above search space design. For instance, LA-loss chooses a scalar $\tau$ and sets $l_i = \tau \log(\boldsymbol{\pi}_i)$. This corresponds to a dictionary containing a single column $\boldsymbol{d} \in \mathbb{R}^K$ with entries $\boldsymbol{d}_i = \log(\boldsymbol{\pi}_i)$.

**Why is train-validation split critical?** In Figure 4 we plot the balanced errors of training/validation/test datasets at each epoch of the search phase. The training data is fixed whereas we evaluate different validation set sizes. The first finding is that training loss always overfits (dotted) until zero error whereas validation loss mildly overfits (dashed vs solid). Secondly, larger validation does help improve test accuracy (compare solid lines). The training behavior is in line with the fact that large capacity networks can perfectly fit and achieve 100% training accuracy [18, 56, 29]. This also means that different accuracy metrics or fairness constraints can be perfectly satisfied. To truly find a model that lies on the Pareto-front of the (accuracy, fairness) tradeoff, the optimization procedure should (approximately) evaluate on the population loss. Thus, as in Figure 4, the validation phase provides this crucial test-proxy in the overpa-

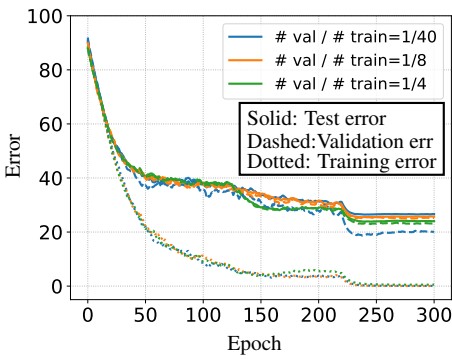

Figure 4: Train/Validation/Test errors during CIFAR10-LT search phase with different validation sizes and fixed training size.

rameterized setting where training error is vacuous. Following the model selection literature [32, 33], the intuition is that, as the dimensionality of the hyper-parameter $\boldsymbol{\alpha}$ is typically smaller than the validation size $n_\mathcal{V}$, validation loss will not overfit and will be indicative of the test even if the training loss is zero. Our search space design in Section 2.2 also helps to this end by increasing the over-sampling ratio $n_\mathcal{V}/\dim(\boldsymbol{\alpha})$ via class clustering. In the appendix, we formalize these intuitions for multi-objective problems (e.g., accuracy + fairness). Under mild assumptions, we show that a small amount of validation data is sufficient to ensure that the Pareto-front of the validation risk uniformly approximates that of the test risk. Concretely, for two objectives $(\mathcal{L}_1, \mathcal{L}_2)$, uniformly over all $\lambda$, the hyperparameter $\boldsymbol{\alpha}$ minimizing the validation risk $\mathcal{L}_{\text{fair}}^{\mathcal{S}_\mathcal{V}}(f) = (1 - \lambda)\mathcal{L}_1^{\mathcal{S}_\mathcal{V}} + \lambda\mathcal{L}_2^{\mathcal{S}_\mathcal{V}}$ in (2.3) also approximately minimizes the test risk $\mathcal{L}_{\text{fair}}(f)$.

## 3 Evaluations for Imbalanced Classes

In this section, we present our experiments on various datasets (CIFAR-10, CIFAR-100, iNaturalist-2018 and ImageNet) when the classes are imbalanced. The goal is to understand whether our bilevel optimization can design effective loss functions that improve balanced error $\mathcal{E}_{\text{bal}}$ on the test set. The setup is as follows. $\mathcal{E}_{\text{bal}}$ is the test objective. The validation loss $\mathcal{L}_{\text{fair}}$ is the balanced cross-entropy

| Method | CIFAR10-LT | CIFAR100-LT | ImageNet-LT | iNaturalist |
|---|---|---|---|---|
| Cross-Entropy | 30.45 | 62.69 | 55.47 | 39.72 |
| LDAM loss  [8] | 26.37 | 59.47 | 54.21 | 35.63 |
| LA loss ($\tau = 1$) [53] | 23.13 | 58.96 | 52.46 | 34.06 |
| CDT loss [67] | **20.73** | 57.26 | 53.47 | 34.46 |
| AutoBalance: $\tau$ of LA loss | 21.82 | 58.68 | 52.39 | 34.19 |
| AutoBalance: $l$ | 23.02 | 58.71 | 52.60 | 34.35 |
| AutoBalance: $\Delta$ | 22.59 | 58.40 | 53.02 | 34.37 |
| AutoBalance: $\Delta \& l$ | 21.39 | 56.84 | 51.74 | 33.41 |
| AutoBalance: $\Delta \& l$, LA init | 21.15 | **56.70** | **50.91** | **33.25** |

Table 1: Evaluations of balanced accuracy on long-tailed data. Algo. 1 with $\Delta \& l$ design space and LA initialization (bottom row) outperforms most of the baselines, across various datasets.

CE$_{\text{bal}}$. We consider various designs for $\ell_{\text{train}}$ such as individually tuning $w, l, \Delta$ and augmentation. We report the average result of 3 random experiments under this setup.

**Datasets.** We follow previous works [53, 13, 8] to construct long-tailed versions of the datasets. Specifically, for a $K$-class dataset, we create a long-tailed dataset by reducing the number of examples per class according to the exponential function $n_i' = n_i \mu^i$, where $n_i$ is the original number of examples for class $i$, $n_i'$ is the new number of examples per class, and $\mu < 1$ is a scaling factor. Then, we define the imbalance factor $\rho = n_0'/n_K'$, which is the ratio of the number of examples in the largest class ($n_0'$) to the smallest class ($n_K'$). For the **CIFAR10-LT** and **CIFAR100-LT** dataset, we construct long-tailed versions of the datasets with imbalance factor $\rho = 100$. **ImageNet-LT** contains 115,846 training examples and 1,000 classes, with imbalance factor $\rho = 256$. **iNaturalist-2018** contains 435,713 images from 8,142 classes, and the imbalance factor is $\rho = 500$. These choices follow that of [53]. For all datasets, we split the long-tailed training set into 80% training and 20% validation during the search phase (Figure 1b).

**Implementation.** In both CIFAR datasets, the lower-level optimization trains a ResNet-32 model with standard mini-batch stochastic gradient decent (SGD) using learning rate 0.1, momentum 0.9, and weight decay $1e-4$, over 300 epochs. The learning rate decays at epochs 220 and 260 with a factor 0.1. The upper-level hyper-parameter optimization computes the hyper-gradients via implicit differentiation. Because the hyper-gradient is mostly meaningful when the network achieves near zero loss (Thm 1 of [46]), we start the validation optimization after 120 epochs of the training optimization, using SGD with initial learning rate 0.05, momentum 0.9, and weight decay $1e-4$, we follow the same learning rate decay at epoch 220 and 260. For CIFAR10-LT, 20 hyper-parameters are trained, corresponding to $l_y$ and $\Delta_y$ of each 10 classes. For CIFAR100-LT, ImageNet-LT, and iNaturalist, we reduce the search space with cluster sizes of 10, 20, and 40 as visualized in Figure 3(a) and as described in Section 2.2. For ImageNet-LT and iNaturalist, following previous work [53], we use ResNet-50 and SGD for the lower and upper optimizations, For the learning rate scheduling, we use cosine scheduling starting with learning rate 0.05, and batch size 128. In searching phase, we conduct 150 epoch training with 40 epoch warm-up before the loss function design starts. For the retraining phase, we train for 90 epochs, which is the same as [53] but only due to the lack of training resources we change the batch size to 128 and adjust initial learning rate accordingly as suggested by [22].

**Personalized Data Augmentation (PDA).** For PDA, we utilize the AutoAugment [12] policy space and apply a bilevel search for the augmentation policy. Our approach follows existing differentiable augmentation strategies (e.g, [24]); however, we train separate policies for each class cluster to ensure that the resulting policies can adjust to class frequencies. Due to space limitations, please see supplementary materials for further details.

**Results and discussion.** We compared our methods with the state-of-the-art long-tail learning methods. Table 1 shows the results of our experiments where the design space is parametric CE (2.1). In the first part of the table, we conduct experiments for three baseline methods: normal CE, LDAM [8] and Logit Adjustment loss with temperature parameter $\tau = 1$ [53]. The latter choice guarantees Fisher consistency. In the second part of the Table 1, we study Algo. 1 with design spaces $l, \Delta$, and $l \& \Delta$. The first version of Algo. 1 in Table 1 tunes the LA loss parameter $\tau$ where $l$ is parameterized by a single scalar $\tau$ as $l_y = \tau \log(\pi_y)$. The next three versions of Algo. 1 consider tuning $l, \Delta, l \& \Delta$ respectively (Figure 3b-d shows the evolution of the $l$ and $\Delta$ parameters during the optimization). Finally, in last version of Algo. 1, the loss design is initialized with LA loss with $\tau = 1$ (rather than balanced CE). The takeaway from these results is that our approach consistently leads to

a superior balanced accuracy objective. That said, tuning the LA loss alone is highly competitive with optimizing $\boldsymbol{\Delta}$ and $\boldsymbol{l}$ alone (in fact, strictly better for CIFAR10-LT, indicating Algo. 1 does not always converge to the optimal design). Importantly, when combining $\boldsymbol{l}\&\boldsymbol{\Delta}$, our algorithm is able to design a better loss function and outperform all rows across all benchmarks. Finally, when the algorithm further is initialized with LA loss, the performance further improves accuracy, demonstrating that warm-starting with good designs improves performance.

In Table 2, we study the benefits of data augmentation, following our intuitions from Lemma 2. We compare to the differentiable augmentation baseline of MADAO [24] which trains a single policy for the full dataset. PDA is a personalized variation of MADAO and leads to noticeable improvement

| Method | CIFAR10-LT | CIFAR100-LT | ImageNet-LT |
|---|---|---|---|
| MADAO [24] | 24.39 | 59.10 | 55.31 |
| AutoBalance: PDA | 22.53 | 58.55 | 54.47 |
| AutoBalance: $\boldsymbol{\Delta}\&\boldsymbol{l}$ | 21.39 | 56.84 | 51.74 |
| AutoBalance: PDA, $\boldsymbol{\Delta}\&\boldsymbol{l}$ | 20.76 | 56.49 | 51.50 |

Table 2: The evaluations on personalized data optimization.

across all benchmarks (most noticeably in CIFAR10-LT). More importantly, the last two lines of the table demonstrates that PDA can be synergistically combined with the parametric CE (2.1) which leads to further improvements, however, we observe that most of the improvement can be attributed to (2.1).

## 4  Approaches and Evaluations for Imbalanced Groups

While Section 3 focuses on the fundamental challenge of balanced error minimization, a more ambitious goal is optimizing generic fairness-seeking objectives. In this section, we study accuracy-fairness tradeoffs by examining the group-imbalanced setting.

✓ **Setting: Imbalanced groups.** For the setting with $G$ groups, dataset is given by $\mathcal{S} = (\boldsymbol{x}_i, y_i, g_i)_{i=1}^n$ where $g_i \in [G]$ is the group-membership. In the fairness literature, groups represent sensitive or protected attributes. For $(\boldsymbol{x}, y, g) \sim \mathcal{D}$, define the group and (class, group) frequencies as

$$\bar{\boldsymbol{\pi}}_j = \mathbb{P}_{\mathcal{D}}(g = j), \quad \text{and} \quad \boldsymbol{\pi}_{k,j} = \mathbb{P}_{\mathcal{D}}(y = k, g = j), \quad \text{for} \quad (k, j) \in [K] \times [G].$$

The group-imbalance occurs when group or (class, group) frequencies differ, i.e., $\max_{j \in [G]} \bar{\boldsymbol{\pi}}_j \gg \min_{j \in [G]} \bar{\boldsymbol{\pi}}_j$ or $\max_{(k,j)} \boldsymbol{\pi}_{k,j} \gg \min_{(k,j)} \boldsymbol{\pi}_{k,j}$. A typical goal is ensuring that the prediction of the model is independent of these attributes. While many fairness metrics exist, in this work, we focus on the Difference of Equal Opportunity (DEO) [23, 17]. Our evaluations also focus on binary classification (with labels denoted via $\pm$) and two groups ($K = G = 2$). With this setup, the DEO risk is defined as $\mathcal{L}_{\text{deo}}(f) = |\mathcal{L}_{+,1}(f) - \mathcal{L}_{+,2}(f)|$. Here $\mathcal{L}_{k,j}(f)$ is the (class, group)-conditional risk evaluated on the conditional distribution of "Class $k$ & Group $j$". When both classes are equally relevant (rather than $y = +1$ implying a semantically positive outcome), we use the symmetric DEO:

$$\mathcal{L}_{\text{deo}}(f) = |\mathcal{L}_{+,1}(f) - \mathcal{L}_{+,2}(f)| + |\mathcal{L}_{-,1}(f) - \mathcal{L}_{-,2}(f)|. \tag{4.1}$$

We will study the pareto-frontiers of the DEO (4.1), group-balanced error, and standard error. Here, group-balanced risk is defined as $\mathcal{L}_{\text{bal}}^{\mathcal{G}}(f) = \frac{1}{KG} \sum_{k=1}^{K} \sum_{j=1}^{G} \mathcal{L}_{k,j}(f)$. Note that this definition treats each (class, group) pair as its own (sub)group. Throughout, we explicitly set the validation loss to cross-entropy for clarity, thus we use CE, $\text{CE}_{\text{bal}}^{\mathcal{G}}$, $\text{CE}_{\text{deo}}$ to refer to $\mathcal{L}$, $\mathcal{L}_{\text{bal}}^{\mathcal{G}}$, $\mathcal{L}_{\text{deo}}$.

**Validation (upper-level) loss function.** In Algo. 1, we set $\mathcal{L}_{\text{fair}} = (1 - \lambda_{\text{val}}) \cdot \text{CE} + \lambda_{\text{val}} \cdot \text{CE}_{\text{deo}}$ for varying $0 \leq \lambda_{\text{val}} \leq 1$. The parameter $\lambda_{\text{val}}$ enables a trade-off between accuracy and fairness. Within $\lambda_{\text{val}}$, we use the subscript "*val*" to highlight the fact that we regularize the validation objective rather than the training objective.

**Group-sensitive training loss design.** As first proposed in [38], the parametric cross-entropy (CE) can be extended to (class, group) imbalance by extending hyper-parameter $\boldsymbol{\alpha}$ to $[K] \times [G]$ variables $\boldsymbol{w}, \boldsymbol{l}, \boldsymbol{\Delta} \in \mathbb{R}^{[K] \times [G]}$ generalizing (2.1), (2.2). This leads us to the following parametric loss function for group-sensitive classification [1]

$$\ell_{\text{train}}(y, g, f(\boldsymbol{x}); \boldsymbol{\alpha}) = -w_{yg} \log \left( \frac{e^{\sigma(\Delta_{yg}) f_y(\boldsymbol{x}) + l_{yg}}}{\sum_{k \in [K]} e^{\sigma(\Delta_{kg}) f_k(\boldsymbol{x}) + l_{kg}}} \right). \tag{4.2}$$

Here, $w_{yg}$ applies weighted CE, while $\Delta_{yg}$ and $l_{yg}$ are logit adjustments for different (class, groups). This loss function is used throughout the imbalanced groups experiments.

---

[1]This is a generalization to multiple classes of the proposal in [38] for binary group-sensitive classificaiton.

| Loss function | Balanced Error | Worst (class, group) error | DEO |
|---|---|---|---|
| Cross entropy (CE) | 23.38 | 43.25 | 33.75 |
| $\mathrm{CE}_{\mathrm{bal}}^{\mathcal{G}}$ | 20.83 | 36.67 | 20.25 |
| Group-LA loss | 22.83 | 40.50 | 29.33 |
| $\mathrm{CE}_{\mathrm{deo}}$ | 19.29 | 35.17 | 25.25 |
| $0.1 \cdot \mathrm{CE} + 0.9 \cdot \mathrm{CE}_{\mathrm{deo}}$ ($\lambda = 0.1$) | 20.06 | 31.67 | 26.25 |
| DRO [58] | 16.47 | 32.67 | 6.91 |
| AutoBalance: $\mathcal{L}_{\mathrm{fair}}$ with $\lambda_{\mathrm{val}} = 0.1$ | **15.13** | **30.33** | **4.25** |

Table 3: Comparison of fairness metrics for group-imbalanced experiments. The first six rows are different training loss choices, where $\mathrm{CE}_{\mathrm{bal}}^{\mathcal{G}}$, Group-LA, $\mathrm{CE}_{\mathrm{deo}}$, and DRO promote group fairness. The last row is Algo. 1, which designs training loss for the validation loss choice of $0.1 \cdot \mathrm{CE} + 0.9 \cdot \mathrm{CE}_{\mathrm{deo}}$. We note that, DEO can be trivially minimized by always predicting the same class. To avoid this, we use a mild amount of CE loss with $\lambda_{\mathrm{val}} = 0.1$ in Algo. 1.

**Baselines.** We will compare Algo. 1 with training loss functions parameterized via $(1-\lambda) \cdot \mathrm{CE} + \lambda \cdot \mathcal{L}_{\mathrm{reg}}$. Here $\mathcal{L}_{\mathrm{reg}}$ is a fairness-promoting regularization. Specifically, as displayed in Table 3 and Figure 5, we will set $\mathcal{L}_{\mathrm{reg}}$ to be $\mathrm{CE}_{\mathrm{bal}}^{\mathcal{G}}$, $\mathrm{CE}_{\mathrm{deo}}$ and Group LA. "Group LA" is a natural generalization of the LA loss to group-sensitive setting; it chooses weights $w_g = 1/\bar{\pi}_g$ to balance group frequencies and then applies logit-adjustment with $\tau = 1$ over the classes conditioned on the group-membership.

**Datasets.** We experiment with the modified Waterbird dataset [58]. The goal is to correctly classify the bird type despite the spurious correlations due to the image background. The distribution of the original data is as follows. The binary classes $k \in \{-, +\}$ correspond to $\{\text{waterbird}, \text{landbird}\}$, and the groups $[G] = \{1, 2\}$ correspond to $\{\text{land background}, \text{water background}\}$. The fraction of data in each (class, group) pair is $\pi_{-,2} = 0.22$, $\pi_{-,1} = 0.012$, $\pi_{+,2} = 0.038$, and $\pi_{+,1} = 0.73$. The landbird on the water background ($\{+, 2\}$) and the waterbird on the land background ($\{-, 1\}$) are minority sub-groups within their respective classes. The test set, following [58], has equally allocated bird types on different backgrounds, i.e., $\pi_{\pm,j} = 0.25$. As the test dataset is balanced, the standard classification error $\mathcal{E}(f)$ is defined to be the weighted error $\mathcal{E}(f) = \pi_{y,g} \mathcal{E}_{y,g}(f)$.

**Implementation.** We follow the feature extraction method from [58], where $x_i$ are 512-dimensional ResNet18 features. When using Algo. 1, we split the original training data into 50% training and 50% validation. The search phase uses 150 epochs of warm up followed by 350 epochs of bilevel optimization. The remaining implementation details are similar to Section 3.

**Results and discussion.** We consider various fairness-related metrics, including the worst (class, group) error, DEO $\mathcal{E}_{\mathrm{deo}}(f)$ and the balanced error $\mathcal{E}_{\mathrm{bal}}^{\mathcal{G}}(f)$. We seek to understand whether AutoBalance algorithm can improve performance on the test set compared to the baseline training loss functions of the form $(1 - \lambda) \cdot \mathrm{CE} + \lambda \cdot \mathcal{L}_{\mathrm{reg}}$. In Figure 5, we show the influence of the parameter $\lambda$ where $\mathcal{L}_{\mathrm{reg}}$ is chosen to be $\mathrm{CE}_{\mathrm{deo}}$, $\mathrm{CE}_{\mathrm{bal}}^{\mathcal{G}}$, or Group-LA (each point on the plot represents a different $\lambda$ value). As we sweep across values of $\lambda$, there arises a tradeoff between standard classification error $\mathcal{E}(f)$ and the fairness metrics. We observe that Algo. 1 significantly Pareto-dominates alternative approaches, for example

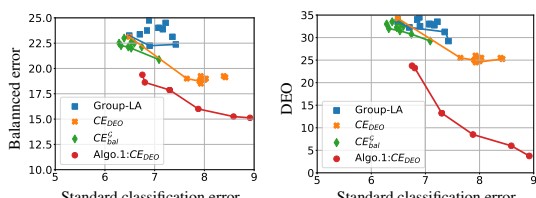

Figure 5: Waterbirds fairness-accuracy tradeoffs for parametrized loss designs $(1 - \lambda) \cdot \mathrm{CE} + \lambda \cdot \mathcal{L}_{\mathrm{reg}}$, for different $\mathcal{L}_{\mathrm{reg}}$ choices. Group-balanced error $\mathcal{E}^{\mathcal{G}}(f)$ (left) and DEO $\mathcal{E}_{\mathrm{deo}}(f)$ (right) are plotted as a function of the misclassification error $\mathcal{E}(f)$. Algo. 1 exhibits a noticeably better tradeoff curve as it uses a DEO-based validation objective to design an optimized training loss function.

achieving lower DEO or balanced error for the same standard error. This demonstrates the value of automatic loss function design for a rich class of fairness-seeking objectives.

Next, in Table 3, we solely focus on optimizing the fairness objectives (rather than standard error). Thus, we compare against $\mathrm{CE}_{\mathrm{deo}}$, $\mathrm{CE}_{\mathrm{bal}}^{\mathcal{G}}$, Group-LA as the baseline approaches as well as blending CE with $\lambda = 0.1$. We also compare to the DRO approach of [58]. Finally, we display the outcome of Algo. 1 with $\lambda_{\mathrm{val}} = 0.1$. While DRO is competitive, similar to Figure 5, our approach outperforms all baselines for all metrics. The improvement is particularly significant when it comes to DEO. Finally, we remark that [38] further proposed combining DRO with the group-adjusted VS-loss for improved performance. We leave the evaluation of AutoBalance for such combinations to future.

# 5 Related Work

Our work relates to imbalanced classification, fairness, bilevel optimization, and data augmentation. Below we focus on the former three and defer the extended discussion to the supplementary.

**Long-tailed learning.** Learning with long-tailed data has received substantial interest historically, with classical methods focusing on designing sampling strategies, such as over- or under-sampling [40, 60, 9, 41, 1, 71, 57, 68, 5, 50]. Several loss re-weighting schemes [54, 52, 25, 13, 6, 34] have been proposed to adjust weights of different classes or samples during training. Another line of work [70, 37, 53] focuses on post-hoc correction. More recently, several works [44, 35, 16, 34, 8, 13, 53, 67] develop more refined class-balanced loss functions (e.g. (2.1)) that better adapt to the training data. In addition, several works [30, 74] point out that separating the representation learning and class balancing can lead to improvements. In this work, our approach is in the vein of class-balanced losses; however, rather than fixing a balanced loss function (e.g. based on the class probabilities in the training dataset), we employ our Algorithm 1 to automatically guide the loss design.

**Group-sensitive and Fair Learning.** Group-sensitive learning aims to ensure fairness in the presence of under-represented groups (e.g., gender, race). [7, 23, 69, 64] propose several fairness metrics as well as insightful methodologies. A line of research [4, 19] optimize the worst-case loss over the test distribution and further applications motivate (label, group) metrics such as equality of opportunity [23, 17] (also recall DEO (4.1)). [58] discusses group-sensitive learning in an over-parameterized regime and proposes that strong regularization ensures fairness. Closer to our work, [58, 38] also study Waterbirds dataset. Compared to the regularization-based approach of [58], we explore a parametric loss design (inspired by [38]) to optimize fairness-risk over validation. [17] proposes methods and statistical guarantees for fair empirical risk minimization. A key observation of our work is that, such guarantees based on training-only optimization can be vacuous in the overparameterized regime. Thus, using train-validation split (e.g. our Algo 1) is critical for optimizing fairness metrics more reliably. This is verified by the effectiveness of our approach in the evaluations of Section 4.

**Bilevel Optimization.** Classical approaches [61] for hyper-parameter optimization are typically based on derivative-free schemes, including random search [65] and reinforcement learning [76, 3, 66, 72]. Recently, a growing line of works focus on differentiable algorithms that are often faster and can scale up to millions of parameters [46, 49, 59, 28, 48]. These techniques [45, 36, 75, 47] have shown significant success in neural architecture search, learning rate scheduling, regularization, etc. They are typically formulated as a bilevel optimization problem: the upper and lower optimizations minimize the validation and training losses, respectively. Some theoretical guarantees (albeit restrictive) are also available [11, 21, 2, 55]. Different from these, our work focuses on principled design of training loss function to optimize fairness-seeking objectives for imbalanced data. Here, a key algorithmic distinction (e.g. compared to architecture search) is that, our loss function design is only used during optimization and not during inference. This leads to a more sophisticated hyper-gradient and necessitates additional measures to ensure stability of our approach (see Algo 1).

# 6 Conclusions and Future Directions

This work provides an optimization-based approach to automatically design loss functions to address imbalanced learning problems. Our algorithm consistently outperforms, or is at least competitive with, the state-of-the-art approaches for optimizing balanced accuracy. Importantly, our approach is not restricted to imbalanced classes or specific objectives, and can achieve good tradeoffs between (accuracy, fairness) on the Pareto frontier. We also provide theoretical insights on certain algorithmic aspects including loss function design, data augmentation, and train-validation split.

**Potential Limitations, Negative Societal Impacts, & Precautions:** Our algorithmic approach can be considered within the realm of automated machine learning literature (AutoML) [27]. AutoML algorithms often optimize the model performance, thus reducing the need for engineering expertise at the expense of increased computational cost and increased carbon footprint. For instance, our procedure is computationally more intensive compared to the theory-inspired loss function prescriptions of [53, 8]. A related limitation is that Algo. 1 can be brittle in extremely imbalanced scenarios with very few samples per class. We took the several steps to help mitigate such issues: first, our algorithm is initialized with a Bayes consistent loss function to provide a warm-start (such as the proposal of [53]). Second, to improve generalization and avoid overfitting to validation, we reduce the hyperparameter search space by grouping the classes with similar frequencies. Finally, evaluations show that the designed loss functions are interpretable (Fig. 3(b,c,d)) and are consistent with theoretical intuitions.

## Acknowledgments

This work is supported in part by the National Science Foundation under grants CCF-2046816, CNS-1932254, CCF-2009030, HDR-1934641, CNS-1942700 and by the Army Research Office under grant W911NF-21-1-0312.

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
