# OpenReview forum: "AutoBalance: Optimized Loss Functions for Imbalanced Data"
_NeurIPS.cc/2021/Conference — NeurIPS 2021 Poster_

### Official Review · Reviewer_oNxf · 2021-07-07

**Rating:** 7
**Confidence:** 3

**Summary:**

This paper studies learning in settings with imbalanced data or under-represented subgroups in the data, where the target objective is fairness-based. E.g. in the setting of imbalanced data, the target objective is to maximize the balanced test accuracy, and in the setting with minority subgroups, target objectives studied are balanced (w.r.t. subgroups) test accuracy, and Difference of Equal Opportunity (DEO) risk.

The paper is motivated by the fact that many existing methods for imbalanced datasets rely on training theoretically-motivated modifications to the cross-entropy loss, where there are multiplicative and additive adjustments applied to the predicted logits for each class. This paper proposes a bi-level optimization procedure to directly learn these modifications, by directly learning the parameters governing the multiplicative and additive adjustments to the class logits. The bi-level optimization optimizes a classifier on a loss defined with specific instantiations of the adjustment parameters in an inner loop, and the outer loop optimizes some desired test objective (e.g., class balanced loss) on a separate holdout validation set. The paper also proposes learning class-specific data augmentation, with the motivation that minority classes are regularized more strongly than majority classes.

The empirical results evaluate the algorithm on imbalanced class and imbalanced subgroup settings.n They show that their algorithms can outperform existing methods in several evaluation metrics.

**Limitations And Societal Impact:**

Yes. Authors address limitations and societal impact in the conclusion section of their paper.

**Main Review:**

In summary, this is a strong paper with clear writing, a clean and well-motivated algorithm, and impressive empirical performance. More details below:

Originality: The methods in this paper are novel and related work is well-cited.

Quality: Claims are well-supported by strong empirical results.

Clarity: The paper is generally clearly written and easy to follow.

Significance: The results in this paper serve as a nice way to encompass some of the more theoretically-motivated prior work which needed to mathematically derive and justify different modifications to the cross entropy loss. In contrast, this paper observes that directly learning these modifications is possible, given the low dimensionality of the parameter space for cross entropy loss adjustments. This is a nice idea and a neat application of AutoML to the imbalanced learning setting, and could hopefully motivate to other applications of AutoML in related scenarios.

Some questions/concerns for the authors:

-- The main concern is that the experiments for the imbalanced group setting are only performed for a single dataset -- it would be nice to see experimental results for additional datasets.
-- What is the comparison of runtime for the proposed algorithm v.s. prior work? Also, what is the schedule for retraining the parameters after the search phase has concluded (e.g, learning rate, # of epochs, etc.)
-- How sensitive are the results to the exact choice of parameters $\alpha$ found by the search phase?
-- For the imbalanced group setting, is there any extension of DEO to the case with multiple groups in a class?

**Time Spent Reviewing:**

3

---

> ### Author Response · Authors · 2021-08-10
> **Response to Reviewer oNxf**
>
> Many thanks for the positive feedback. We address the comments & questions below.
>
> **(1) Re: “More experiments for imbalanced groups.”** We agree that more experiments can make a stronger case. To address this, we ran further experiments on the CelebA dataset (following [Sagawa et al. ICLR’20]) where we use “hair color” as the label and “gender” as the group membership. This dataset contains 202599 examples in total, of which we kept 162770 training samples with 1387 in the smallest group. We used the same evaluation procedure using the baselines in Table 8 of the supplementary materials. In short, the results are consistent with the Waterbird experiments. Algo. 1 outperforms Sagawa et al.’s DRO by at least 1% in all three performance metrics. Performance is even better compared to other baseline methods. We will add the CelebA results in the final version. We also note that the DRO row from Table 8 will be moved to Table 4 in the final manuscript.
>
> **(2) Re: “Runtime comparison & Retraining schedule”** Our procedure requires two training phases, each using the whole dataset (search phase and retraining). So intuitively, the overall cost is at least twice that of (standard) training with a fixed loss function. For our CIFAR100-LT experiments (optimizing both $\Delta$ & $l$), we found that the search phase takes 3 times as long as retraining, so the overall cost is 4 times that of standard training on a single RTX 2080 Ti GPU, which is a fairly favorable runtime. In the revised version, we will also report the runtime on ImageNet-LT and iNaturalist and provide a discussion on the impact of the number of loss function parameters on the runtime. While we are not aware of comparable prior work, following the feedback of Reviewer mSPX, we ran experiments on Bayesian optimization (specifically SMAC method of [Hutter et al. LION'11]) for hyperparameter tuning and we found that our differentiable approach is noticeably faster under a reasonable evaluation setting (see the last item of our response to Reviewer mSPX). Retraining phase is very similar to the search phase, but uses a different number of epochs and learning rate schedule. This is to ensure we keep the same setup as [8,50] to make a fair comparison, in which we train for 200 epochs, the batch size is 128 and weight decay is 1e-4. The base learning rate is 0.1 and it is reduced by a factor of 10 at epochs 160 and 180. For ImageNet-LT and iNaturalist, we train for 90 epochs with learning rate 0.4 using cosine learning rate decay again following [8,50].
>
>
> **(3) Re: “Sensitivity to the exact choice.”** Observe that the train-validation split of the search phase introduces stochasticity. Thus, the fact that the loss function found by the search phase works well for the retraining phase indicates that the loss function choice is reasonably robust to the exact choice. We will do further investigation to assess the degree of variability of the loss function parameters as we generate different train-validation splits (i.e. different random partitionings of the overall dataset).
>
> **(4) Re: “Multi-group DEO.”** We were not able to find a reference for multi-group variation of DEO, that is why we didn’t provide a definition. Most applications seem to involve the two group scenario e.g. [Sagawa et al. ICLR’20]. That said, a natural extension to multi-group setting can be the L1 distance from the mean error over groups. Focusing on the + label, this would be  $\sum_{g=1}^G |L_{+,g} - \frac{1}{G}\sum_{g’=1}^G L_{+,g’}|$. We will add this discussion to the paper.

---

> > ### Comment · Reviewer_oNxf · 2021-08-30
> > **Thanks for the response**
> >
> > Thanks for answering the questions in my review. As reviewer mSPX raised some concerns about overfitting to the validation set, I'd like to ask a clarifying question regarding how the error bars are computed (e.g. for Table 5, or for the recent response to reviewer mSPX). Do these error bars average over the train-val split of the dataset, or is this always fixed over all the trials? Given reviewer mSPX's concerns about overfitting to the validation set when the sample size for rare classes is very small, I think it would be ideal to see error bars which re-randomize the train-val split each trial.

---

> > > ### Author Response · Authors · 2021-08-30
> > > **Computation of error bars**
> > >
> > > Dear Reviewer,
> > >
> > > Thank you for the follow-up question. In short, the tables we posted in response to Reviewer mSPX uses “three runs with independent train-validation splits”. However, the experimental results in the paper and in Table 5 use “independent runs (e.g. SGD and random initialization of the network) on the same split”. Also we report the standard error $\frac{\sigma}{\sqrt{\text{num of trials}}}$ rather than standard deviation $\frac{\sigma}{\sqrt{\text{num of trials - 1}}}$.
> > >
> > > As the reviewer already points out, the reason we particularly used “independent splits” for Reviewer mSPX is because of the sensitivity of the validation error of the tail classes due to small sample size. Specifically, if we used the same split, in the tables we posted, we would see a larger bias and smaller standard error in the gap “validation error - test error”.  Intuition is that, if the split picks an “easy” validation example for a tail class, in all three experiments we will consistently achieve zero validation error. On this tail class, we would have zero standard error but significant bias.
> > >
> > > For balanced test error, potential bias due to same the split is less of a concern, because we average over all classes and the test standard error is significantly smaller than the validation standard error. The latter can be seen from the table below which is a subset of Tables B and D posted to Reviewer mSPX; however, this new table only reports the overall “Balanced Errors” on CIFAR-10&100-LT. We also numerically checked that “same split” does not significantly affect the average balanced test errors compared to “independent splits” (note that in expectation over random splits, they are the same).
> > >
> > > | Balanced err                | CIFAR-10            | CIFAR-100            |
> > > |------------------|------------------|------------------|
> > > | Training error     | 0.34$\pm$0.00 | 0.90$\pm$0.08 |
> > > | Validation error | 22.24$\pm$1.66 | 61.15$\pm$1.45 |
> > > | Test error      | 23.49$\pm$0.26 | 61.21$\pm$0.16 |
> > >
> > > Please let us know if you have further questions or suggestions. Thank you.

---

> > > > ### Comment · Reviewer_oNxf · 2021-08-31
> > > > **Thanks for the clarification.**
> > > >
> > > > Thanks for the clarification! This addresses my concern. I'm keeping my score the same.

---

### Official Review · Reviewer_GFgA · 2021-07-15

**Rating:** 7
**Confidence:** 4

**Summary:**

This work introduces a novel algorithm (AutoBalance) to address the imbalance and group sensitive classification problems and maximise the fairness-seeking objectives. The algorithm uses a bi-level optimization framework to optimize the model weights over the training data while automatically fine-tuning the loss function by monitoring the validation data. In addition, AutoBalance uses data augmentation policies personalised to subpopulations in the data.

**Limitations And Societal Impact:**

The paper adequately and fairly outlines its limitations in a dedicated section.

**Main Review:**

\+ means strength, \- means weakness

**Originality**

\+ *The paper presents a novel algorithm.* The algorithms applies ideas from bilevel optimization [43, 46, 55, 26, 45] to fairness and class imbalance learning. Unlike previous work [41, 33, 16, 32, 8, 13, 50, 63], which employs fixed class balancing loss functions based on some training data statistics, the proposed method automatically guides the loss function design and learns a more optimal set of parameters for the balanced loss.

**Quality**

\+ *Claims are well supported through experimental evidence.*

\+ *The authors are careful and honest about reporting the paper's limitations.*

\+ *The paper compares its method to recent state-of-the-art [8,50] and beats them.*

\- *Lack of more baselines.* The paper compares performance to only two baselines [8,50], and it is not clear to me why more baselines are not evaluated (e.g. [41, 33, 16, 32, 13, 63]).

\- *No error bars in the results.* There are no error bars in the result tables, making it more challenging to assess the significance of the improvements.

**Clarity**

\+ *The paper is written very well, and it is easy to follow.* On the most part, the paper adequately informs the reader.

\- *Some clarifications about the experiments are needed.* Table 1 does not have any $PDA$ component in Algo. 1, does this mean that no $PDA$ was applied in those experiments? (if $PDA$ was used, did the authors also apply it to the baselines?) Similarly, did experiments in Table 2 apply $LA\~\text{init}$? And lastly, does Table 3 show results for "Algo 1. $ \alpha \leftarrow \Delta \\& l, LA~\text{init}, PDA$"?

\- *A possible inconsistency in the results* but I may also be misunderstanding something (see above for my confusion). Specifically, if I am correct in understanding, the diagonal results in Table 3 show the performance for "Algo 1. $ \alpha \leftarrow \Delta \\& l, LA\~\text{init}, PDA$" which have a higher error rate than both "Algo 1. $ \alpha \leftarrow PDA, \Delta \\& l $" in Table 2 and "Algo 1. $ \alpha \leftarrow \Delta \\& l, LA\~\text{init}$ " in Table 1. This suggests that, in fact, $PDA$ and $LA\~\text{init}$ degrade performance when applied together. If my understanding is correct, this undermines the second contribution point (lines 73-77) and should be explicitly clarified in the paper. If my understanding is wrong, why are the results in the diagonal in Table 3 different from either the bottom row of Table 1 or Table 2?

**Significance**

\+ *This work aims to address a timely problem of fairness-seeking optimization, and the paper will be relevant to the community.*

___
### Post rebuttal update
___
I have read other reviewers' comments and the authors' rebuttals, and I am inclined to keep my original score of 7 with increased confidence of 4, condioned on the additional baselines being included in Table 1 in the main paper. The rebuttal addresses the clarity issues, and the additional results show that the method is at least as good as other baselines [28][63][41] in the Cifar-10-LT setting. The performance advantage over [28][63] in the ImageNet-LT dataset is much more obvious. Although results for [41] in the ImageNet-LT setting were not provided, likely the method would also be beaten like in the Cifar-10-LT setting. I think this work is good, addressing a timely problem and it will be relevant for the NeurIPS conference.

**Time Spent Reviewing:**

6

---

> ### Author Response · Authors · 2021-08-10
> **Response to Reviewer GFgA**
>
> We greatly appreciate your positive review and helpful feedback.
>
> **(1) Re. “Lack of more baselines”** We agree with this suggestion. We will enlarge Table 1 to incorporate three or more baselines including [28, 41, 63] which use similar experiments. In the supplementary material, we will also augment Table 5 and make it more comprehensive. We also note that [8] demonstrates that LDAM is consistently better than [41] and [50] demonstrates that logit-adjustment is consistently better than [28]. Since we compare against and consistently outperform [8,50], it seems safe to deduce that, our approach also outperforms these.
>
> **(2) Re. “No error bars in the results”** The error bars are currently provided in Tables 5-8 of the supplementary material. To address reviewer feedback, we will add error bars to Tables 1&4. However we prefer to keep Tables 2&3 intact to avoid clutter, as their font size is smaller.
>
> **(3) Re. “Clarifications on experiments”** That is correct. Table 1 is dedicated to optimizing the parametric cross-entropy and does not consider PDA. The experiments in Table 2 do not apply LA init. Table 3 shows results for “tuning $\Delta$ & $l$” which is the second to last line of Table 1. However, based on reviewer feedback, we agree that the paper can benefit from evaluating the combined approach of “$\Delta$ & $l$, LA init, PDA”. We will update Table 2 and 3 accordingly in the final version (or at least provide these evaluations in the supplementary). We will also revise the captions of Tables 2&3 to clarify the setting.
>
> **(4) Re. “A possible inconsistency in the results”** As mentioned above, the diagonal results in Table 3 correspond to the second to last line of Table 1. Table 3 is not using PDA (Table 2) or LA initialization (last line of Table 1). We will clarify this in the caption.

---

> > ### Comment · Reviewer_GFgA · 2021-08-24
> > **Thanks for the response, but it would be useful see the results for the suggested baselines.**
> >
> > I thank the authors for the clarifications and the rebuttal to other reviewer's comments. I would be more convinced if I could see the actual results for [28,41,63] as suggested by the authors before I make my final decision.

---

> > > ### Author Response · Authors · 2021-08-30
> > > **Update**
> > >
> > > Dear Reviewer,
> > >
> > > Thank you for your comment and query. We are still in the process of running these experiments. We will post a table of what results we have by Tuesday, August 31 at the latest, to avoid further delay. We appreciate your patience.
> > >
> > > Authors

---

> > > ### Author Response · Authors · 2021-09-01
> > > **Comparisons to [28,41,63] (new version includes ImageNet-LT)**
> > >
> > > Dear Reviewer,
> > >
> > > Thank you again for your inquiry and your patience. Below, we report the outcomes of the comparisons to [28,41,63]. We ran all the baselines using the same experimental configurations described in the paper to ensure fair comparison. All new experiments from [28,41,63] are averages of three runs for the CIFAR-10-LT and CIFAR-100-LT datasets. We are still in the process of running ImageNet and will update as we get further results. Below we display the balanced errors.
> > >
> > > | Method                       	| CIFAR-10-LT | CIFAR-100-LT | ImageNet-LT |
> > > |----------------------------------|-------------|--------------|----------|
> > > | Focal loss [41]               	| 29.66 $\pm$ 0.21| 61.63 $\pm$ 0.24| TBD |
> > > | CDT loss [63]                 	| 20.73 $\pm$ 0.27| 57.26 $\pm$ 0.29|    57.18 $\pm$ 0.2   |
> > > | $\tau$-normalization  [28]          |  22.05$\pm$1.43      |  56.89$\pm$0.51       |   57.42 $\pm$ 0.13  |
> > > | LWS [28]                          |  29.03$\pm$0.87       |  58.96$\pm$0.53       |  57.3 $\pm$ 0.57   |
> > > | $\alpha\gets\Delta+l$ | 21.39 $\pm$  0.18 | 56.84 $\pm$ 0.17 |  53.16 $\pm$ 0.17   |
> > > | $\alpha\gets\Delta+l+\text{PDA}$ | 20.69 $\pm$ 0.17 | 56.47 $\pm$ 0.23 |  52.18 $\pm$ 0.22   |
> > >
> > > **Experimental details:** Here the last two lines are taken from Tables 5 and 6 of the supplementary material. These entries report the average over 5 runs, and the standard errors (which will be smaller than over 3 runs). [63] uses the multiplicative adjustment term $\Delta$ and chooses $\Delta_y$ based on class frequency as $(\pi_y/\pi_{\max})^\gamma$. $\gamma$ is the tuning parameter. In our implementation of [63], as recommended in their paper, we searched $\gamma$ over the $[0,0.5]$, $[0,0.2]$, and $[0,0.2]$ intervals for CIFAR-10, CIFAR-100, and ImageNet respectively. For CIFAR, we used a grid search with spacings 0.1 and 0.05, which led to the choices $\gamma=0.4$ and $\gamma=0.15$, respectively. For ImageNet, we used a finer grid of ${0.02,0.05,0.75,0.1,0.15,0.2}$ which led to the choice $0.75$. We note that CIFAR-10-LT results are consistent with what they report, however we were not able to reproduce their CIFAR-100-LT results despite using their suggested search interval [0,0.2]. We suspect there are some differences between experimental configurations. [63] does not report ImageNet. [28] proposes $\tau$ normalization, and LWS is a variation of it. These are posthoc adjustment methods that scale the rows of the softmax layer. For CIFAR-10-LT, we found that LWS does not work well. The reason (as far as we understand) is that LWS uses a balanced subsample from the training data and treats it as a validation to learn its hyper-parameters. However, CIFAR-10-LT training overfits so LWS cannot learn useful hyper-parameters. Finally, we always tuned $\tau$-normalization over the grid $[0,2]$ with spacing $0.1$ suggested by the authors. For all datasets, optimal $\tau$ was strictly within $[0.7,1.9]$. For ImageNet-LT, the $\tau$ choices were $0.7,0.7,0.8$ in our three runs. For ImageNet-LT, our experiments reported above use ResNet-50 with batch-size 128 whereas [28] used batch-size 512 and [63] did not report ImageNet-LT. Table 7 of [28] provides ResNet-50 results which are slightly better (56.6 and 57.1 for $\tau$-normalization and LWS respectively). In any case, there is a 3% gap between our method and [28,63]. We did not prioritize Focal loss for ImageNet as its performance on CIFAR is lackluster.
> > >
> > > We should remark that $\tau$ of [28] and $\gamma$ of [63] is tuned on the test dataset in the interest of time. This might give them a slight edge. Also see our Comment 4/4 to Reviewer mSPX regarding the practice of tuning on test data in related works [8,36,50,63].
> > >
> > > **Discussion:** Based on these experiments, we see again that our approaches generally outperform these baselines in both datasets. The benefit of our approach is most visible in the ImageNet-LT benchmark (which is arguably a more insightful baseline compared to CIFAR-LT due to containing many more classes and higher resolution images). Among the baselines $\tau$-normalization achieves good results on CIFAR-100-LT. However, we found that it has higher variability (e.g. standard error). CDT performs well on CIFAR-10-LT and deserves its own discussion. Note that our design space includes the $\Delta$-adjustment of CDT. The fact that CDT can perform better than $\Delta+l$ (even for CIFAR-10) highlights the optimization challenge. Our optimization process does not incorporate any prior information and learns the roles of $\Delta$&$l$ from scratch (e.g. see Fig 3(b,c,d)). Instead, by setting $\Delta_y=(\pi_y/\pi_{\max})^\gamma$, CDT explicitly incorporates the prior that majority classes should get larger $\Delta_y$. We believe fusing such prior information can indeed regularize the optimization and help discover better hyper-parameters. In support of this claim, the last two lines of Table 1 demonstrate how infusing “LA init” can consistently improve over $\alpha\gets\Delta+l$ (e.g. 1% ImageNet improvement). Finally, as we mentioned to Reviewer mSPX, both our “class clustering” and the choices provided by LDAM [8], CDT [63], LA [50] losses can be unified under a “search space design” idea. The idea is infusing prior knowledge into optimization via a tall dictionary matrix $D$. $D$ determines the search space by representing high-dimensional $\Delta$&$l$ as $\Delta=D\Delta’$ and $l=Dl’$ where $\Delta’$&$l’$ are lower-dimensional embeddings to optimize. The design choices of [8,50,63] and our clustering can be all mapped to specific dictionaries (e.g. polynomials/logarithm of $\pi_y$, cluster-membership). In the revised version, we plan to motivate our clustering strategy via this “search space design” viewpoint.
> > >
> > > Finally, the revised manuscript will expand our Table 1 with [28,41,63] evaluated also on ImageNet (ongoing) and iNaturalist. We would be happy to address further questions.
> > >
> > > Thank you,
> > >
> > > Authors

---

> > > > ### Comment · Reviewer_GFgA · 2021-09-01
> > > > **Thanks for the additional baselines**
> > > >
> > > > Thank you for the results of additional baselines showing that the proposed methods perform at least as well as the other baselines, although not by a significant margin.

---

> > > > > ### Author Response · Authors · 2021-09-05
> > > > > **ImageNet-LT comparison to [28,63]**
> > > > >
> > > > > Dear Reviewer,
> > > > >
> > > > > We wanted to follow-up with the ImageNet-LT comparison since it is a larger-scale benchmark compared to CIFAR-LT. The details are provided within our earlier post. To summarize, we found that our approach has a clearer benefit for ImageNet-LT compared to [28,63].
> > > > >
> > > > > | Method           |            ImageNet-LT (Balanced Error) |
> > > > > |--------------------|---------------------------------------------------|
> > > > > | CDT loss [63] |                                57.18 $\pm$ 0.20 |
> > > > > | $\tau$-normalization [28] |              57.42 $\pm$ 0.13 |
> > > > > | LWS [28] |                                       57.30 $\pm$ 0.57 |
> > > > > | $\alpha\gets\Delta+l$ |                   53.16 $\pm$ 0.17 |
> > > > > | $\alpha\gets\Delta+l+\text{PDA}$ | 52.18 $\pm$ 0.22 |
> > > > >
> > > > > This unfortunately took a bit of time as we wanted to avoid inaccurate/unfair comparisons. Specifically, the ImageNet-LT results we report use ResNet-50 with batch-size 128 whereas [28] used batch-size 512 and [63] did not report ImageNet-LT. Table 7 of [28] provides ResNet-50 results which are slightly better (56.6 and 57.1 for $\tau$-normalization and LWS respectively). In any case, there is a 3% gap between our method and [28,63].
> > > > >
> > > > > We understand that the review period may already be over and we thank you again for your patience.

---

### Official Review · Reviewer_WkxY · 2021-07-18

**Rating:** 7
**Confidence:** 3

**Summary:**

The paper demonstrates how to jointly optimize parametrized cross entropy and model parameters and shows improvement on datasets with class imbalance.

**Limitations And Societal Impact:**

1. Paper contributions state that it "automatically designs a training loss function". However the actual training loss is cross entropy only with multiplicative and additive parameters. Since the actual loss is a variation of bias+weighted cross entropy this needs to be mentioned in both abstract and contributions in clear terms. There are other works out there which use different kind of loss functions to optimize for class imbalance [1]- they should be cited.
2. The papers is written in a complicated way and is difficult to read and understand. Simple statements have been made into complex math jargon, and a lot of places terms are first used in equation and then defined later.
3. Imbalanced data of only images is considered - other modalities should be considered
4. There are many small and large datasets which are not modified to create LT datasets - for example cubs and flowers [1]. The paper would greatly benefit by analyzing performance on such datasets.
5. "Hyper" gradient computation on L_up is critical to implementation of paper - this needs to appear in main text not just supplementary. Its also not completely clear how to implement it.
6. Paper uses data augmentation (previously proposed epsilon ball perturbations - applied to low population classes). Upon first inspection the data augmentation seems to create a bigger impact than the primary algorithm. This needs to be highlighted properly.


[1] Dubey, A., Gupta, O., Guo, P., Raskar, R., Farrell, R., & Naik, N. (2018). Pairwise confusion for fine-grained visual classification. In Proceedings of the European conference on computer vision (ECCV) (pp. 70-86).

**Main Review:**

The central theme of paper is to optimize the model parameters on training data and parametrized cross entropy on validation set. The model itself can be optimized using simple gradient based approaches, more sophisticated approaches from literature are used for computing parametrized cross entropy on the validation set. Additionally data augmentation for minority samples is proposed to increase classifier margin from low population samples. Authors perform extensive experimentation on the cifar, imagenet and inaturalist imbalanced datasets.

**Time Spent Reviewing:**

3

---

> ### Author Response · Authors · 2021-08-10
> **Response to Reviewer WkxY**
>
> We greatly appreciate your thorough review and constructive feedback. We address the comments point-by-point.
>
> **(1) Re: “Training loss is cross-entropy”.** We agree with the reviewer and will revise the first contribution to clarify that loss function design space is parametric cross-entropy functions. We will also add a remark on alternative design options and refer to [1, Dubey et al.].
>
> **(2) Re: “Writing is over-complicated, math jargon”** We agree that writing can be more crisp. In fact, Reviewer FVKf provided related feedback. Please see their comments and our corresponding responses (items 2, 4, 5, 6 of our response). Additionally, in a revised version, we plan to reduce/remove the jargon on bilevel optimization (lower-level, upper-level problems) and stick to training and validation problems to improve clarity.
>
> **(3&4) Re: “only image modality is considered, evaluations on novel datasets such as CUB & Flowers”.** We agree that the paper can benefit from further experiments. Thank you for providing these references. Our priority would be evaluations on another modality (specifically natural language processing problems). A good candidate is the Multi-Genre Natural Language Inference (MultiNLI) dataset by [Williams et al. ACL’18]. This was studied by the related work [Sagawa et al.] which focuses on the group-sensitive setting. We are in the process of starting these experiments and we plan to incorporate them in the revised paper. As a related note, the Waterbird dataset in Sec 6 is formed from CUB & Places datasets.*
>
> **(5) “Add hyper-gradient computation to main text”.** Good point. Currently, hyper-gradient computation (via approximate implicit differentiation) is provided in Algorithm 2 in the supplementary, which follows the recent work [Lorraine et al. AISTATS’20, Rajeswaran et al. NeurIPS'19]. We plan to use the additional 1-page space (provided for accepted papers) to include the algorithm and we will also provide further details on the implementation and the importance of approximate Hessian inverse.
>
> [Lorraine et al.] "Optimizing millions of hyperparameters by implicit differentiation." AISTATS, 2020.
>
> [Rajeswaran et al.] "Meta-learning with implicit gradients." NeurIPS, 2019.
>
>
> **(6) “highlighting the impact of data augmentation”.** Table 2 summarizes our experiments on data augmentation. We observe that the individual benefit of PDA is weaker than the benefit of “tuning $\Delta$ & $l$”. This can be seen by comparing the second line of Table 2 with the second to last line of Table 1. We note that applying PDA on top of “tuning $\Delta$ & $l$” further helps since the last line of Table 2 is uniformly better than “tuning $\Delta$ & $l$”. However, it seems safe to attribute most of the improvement to “tuning $\Delta$ & $l$”, as the reviewer pointed out. We will add a discussion on this.

---

> > ### Comment · Reviewer_WkxY · 2021-08-25
> > **thanks**
> >
> > thanks for your responses

---

### Official Review · Reviewer_mSPX · 2021-07-22

**Rating:** 4
**Confidence:** 4

**Summary:**

This paper proposes a method for long-tailed classification. The key idea is to use a generalized loss function parameterized by hyper-parameters \alpha and to (automatically) tune \alpha on a validation set. Results are presented on CIFAR-10-LT, CIFAR-100-LT, ImageNet-LT and iNaturalist datasets.

**Ethical Concerns:**

No ethical concerns.

**Limitations And Societal Impact:**

Authors have adequately addressed the limitations and potential negative societal impact.

**Main Review:**

Strengths
-------------
(1) Experimental results are good.
(2) The paper is written well and the key technical details are sufficiently described.

Weaknesses
-----------------
(1) Unfortunately, there is limited scientific novelty in this work in my opinion. The main loss function is a straightforward generalization of the loss functions developed in previous works such as [8], [50] and others. The authors then optimize the hyper-parameters of this function on a validation dataset using the method proposed in [43]. Therefore, the overall method proposed in this paper is a highly-engineered solution with limited scientific novelty/interest.

(2) The idea of fitting hyper-parameters to a validation set depends on the premise that (as stated in line 62): "unlike training data, the validation data is difficult to fit and will provide a consistent estimator of the test objective". However, this assumption is flawed. With millions of parameters in the actual model and a handful of hyper-parameters, it is easy to overfit to the validation set as well, specially when the validation set is small. For example, in CIFAR-100-LT, the validation set authors use has only 1 example for the tail-most class (= 500 / 100 * 20%). It is not reasonable to assume that it is difficult to overfit to such a small dataset when automatically optimizing over hyper-parameters using a gradient descent procedure.

(3) Since the loss function itself (Eq. 4.2) is not novel, one could argue that the key contribution of this paper is automatically tuning hyper-parameters on the validation set. But then the method should be compared to other automatic hyper-parameter tuning methods such as Bayesian Optimization methods.

**Time Spent Reviewing:**

3 hours

---

> ### Author Response · Authors · 2021-08-10
> **Response to Reviewer mSPX**
>
> We greatly appreciate your feedback, and it has helped us revise the paper as we explain below. As the reviewer suggests in their comment on “small validation set of long-tail data”, efficient and automated discovery of optimized loss functions presents new technical challenges, which are addressed by the algorithmic novelties of our work. Our key contribution is providing a computationally-efficient and robust algorithm for automated loss function design. (Even if one used other hyperparameter optimization techniques, to the best of our knowledge, this is the first work to propose the algorithmic design of loss functions and verify its empirical benefits over point design choices from earlier works [8, 28, 50, 63].) We start by addressing the second comment on “overfitting validation” which connects well to the first comment on novelties. We then discuss the third comment on comparisons.
>
> **(2) Re: ``With millions of parameters and a handful of hyper-parameters, it is easy to overfit to the validation set.’’** Both the theoretical literature and empirical evidence demonstrate that it is difficult to overfit to the validation set when the number of hyper-parameters is much less than the validation size. Empirically, consider the “Algo. 1: $\alpha <- \Delta$ & $l$” row in our Table 1. After the Search phase (before the Retraining phase in Fig 1b), we obtain the following balanced errors: For CIFAR10-LT: Training 5.06%, Validation 21.72%, Test: 21.88%. For CIFAR100-LT: Training: 2.46%, Validation: 48.52%, Test: 60.67%. Observe that the training errors are indeed close to zero whereas the validation and test errors are fairly close to each other and are much larger than zero. We will add a discussion on this in the revised version and will also add a table of train/validation/test (after the search phase) in the supplementary material.
>
> Theoretically, the model selection literature advocates that the level of overfitting for the validation set is dictated by the # of hyperparameters rather than the # of weights (e.g. Theorem 4 of Kearns et al.). In essence, if the hyperparameter space $\cal{H}$ is discrete, applying a standard union bound via Hoeffding inequality, one only needs $\cal{O}(\log(|H|))$ validation samples. Closer to our setting, our Theorem 1 in the supplementary material establishes a similar result where we allow for continuous hyperparameters and arbitrary linear combinations of multiple test objectives (e.g. blending accuracy and fairness objectives). In short, if the training algorithm is stable (Assumption 1), regardless of the # of training weights, up to logarithmic factors, the required validation sample size grows proportional to $h+R$, where $h$ is the # of hyperparameters and $R$ is the # of objectives to blend. In our work $h$ is at most in the order of number of classes whereas R is at most 2, hence it is difficult to overfit to the validation set. Also see below for our approach to reduce $h$. We also noticed a typo in the right handside of (C.1): $\arg\min \cal{L}_ \lambda (f_{\hat{\alpha}})$ should be $\min \cal{L}_ \lambda(f_{\alpha})$. The proof uses a covering argument to establish uniform convergence of the validation risk to test risk.
>
>
>
> [Kearns et al.] “An Experimental and Theoretical Comparison of Model Selection Methods” Machine Learning, 1997.
>
>
>
> We absolutely agree with the reviewer that long-tail data presents challenges as the sample size per class can be very small. This brings us to the reviewer’s next comment.
>
>
>
> **(1) Re: ``Limited scientific novelty.’’** A small sample size of minority classes presents nontrivial technical challenges, which are addressed by the following algorithmic novelties of the paper: (1) While multiplicative and additive logit adjustments are proposed by recent works, this is the first work (to the best of our knowledge) to demonstrate that combining both can lead to substantial benefits. (2) Unlike earlier works, we demonstrate that small classes can provably benefit from stronger data augmentation, we propose “personalized augmentation”, and we verify its empirical benefits (e.g. Table 2). Note that data augmentation also helps mitigate the “small sample size” concern by increasing the “effective” sample size. (3) To further address “small sample size”, we also cluster the classes and use the same logit adjustment for classes in the same cluster (see Figure 3a). This is done for all datasets except CIFAR-10 (which has more samples per class). This procedure reduces the dimensionality of the hyperparameter search space and increases the “sample size per hyperparameter”. To further address the reviewer’s feedback and to further expand on our technical contribution, we plan to revise the paper and generalize this idea in terms of “search space design”. Designing a good search space is known to be a critical aspect of hyperparameter search problems (see Nas-Bench-101 & 201 papers for the Neural Architecture Search problem). At a high level, the clustering we discuss in item (3) above is a specific point in the space of possibility prior information types (e.g. class frequency, class semantic similarity) -- this search space can be made more general and unify other approaches in the literature. Specifically, the search space can be represented as the range space of a dictionary $D\in \mathbb{R}^{K\times d}$ where $d$ is the dimension of the search space and we wish to ensure $d$ is smaller than the # of classes $K$. We represent hyperparameters via $l=D l’$ and $\Delta=D \Delta’$ where $l’,\Delta’\in \mathbb{R}^d$ are dimensionality-reduced parameterizations of $l,\Delta$. In our work, the columns of the dictionary $D$ correspond to individual clusters that are 1 for classes within the cluster and zero otherwise. This viewpoint also motivates further research as one can design better dictionaries $D$. In fact, approaches in [8,50,63] can be viewed as a dictionary with a single column ($d=1$), where the entries are a polynomial or logarithmic function of the class frequency. For instance, the LA loss of [8] chooses $D_i= \log(\pi_i)$ and writes $l=D \tau$. This yields the fourth row of Table 1 (when tuned with Algo 1). (4) These novelties are complemented by additional refinements (e.g. warm-up phase to help optimization, using weighted validation to ensure unbiased risk), theoretical intuitions/guarantees and further results for imbalanced groups. Also see Reviewer FVKf’s comment on the challenges surrounding bilevel optimization (e.g. hysteric effects).
>
>
>
> **(3) Re: “compare to other methods”.** This is a good suggestion, thank you. To clarify, as mentioned in Line 109, we use differentiable bilevel optimization because our hyperparameter space is continuous and potentially large (e.g. in the order of $K$). This is typically the setting where differentiable optimization can lead to speedup over alternatives. For instance, in neural architecture search (which has >100 hyperparameters), differentiable methods led to significant search cost reduction (e.g. DARTS, FBNet, ProxylessNAS). That said, we agree that the paper can benefit from additional comparisons. We ran experiments with Bayesian optimization (BO), specifically the SMAC method of [Hutter et al. LION'11], for CIFAR10-LT datasets. We verified that BO can also discover competitive loss functions given enough time. Specifically, for CIFAR10-LT, given enough time, our preliminary BO experiment achieves test error of 22.51% compared to 21.39% of our method. However, our experiments indicate that BO takes significantly longer compared to our method. Specifically, our method typically takes 4x as long as standard training (with fixed loss function) whereas, in preliminary experiments, BO takes 10~20 times as long. We suspect this is because BO requires more training time to explore/try different configurations, whereas differentiable bilevel optimization conducts an end-to-end optimization (in a single run). We will provide a more thorough discussion in the revised manuscript (both accuracy and search time) and, besides BO, we will also include a comparison against Hyperband [Li et al. JMLR’17].
>
> [Hutter et al. LION'11] "Sequential model-based optimization for general algorithm configuration", International Conference on Learning and Intelligent Optimization 2011.

---

> > ### Comment · Reviewer_mSPX · 2021-08-26
> > **Response after the rebuttal**
> >
> > I am acknowledging that I have carefully read the other reviews and the author response.
> >
> > I believe any machine learning researcher/practitioner would agree that overfitting to the validation set is a real issue in practice. As an analogy, there are theoretical results saying that MLPs are universal approximators. However, in practice, it is quite difficult to get an MLP to learn a complex decision function on a real dataset (otherwise we would not need specialized architectures like CNNs, Transformers, etc.). Example count for different classes in CIFAR-100-LT validation set is as follows (after doing a ceil() operation):
> > [100.,  96.,  91.,  87.,  83.,  80.,  76.,  73.,  69.,  66.,  63.,
> >         60.,  58.,  55.,  52.,  50.,  48.,  46.,  44.,  42.,  40.,  38.,
> >         36.,  35.,  33.,  32.,  30.,  29.,  27.,  26.,  25.,  24.,  23.,
> >         22.,  21.,  20.,  19.,  18.,  17.,  17.,  16.,  15.,  14.,  14.,
> >         13.,  13.,  12.,  12.,  11.,  11.,  10.,  10.,   9.,   9.,   8.,
> >          8.,   8.,   7.,   7.,   7.,   6.,   6.,   6.,   6.,   5.,   5.,
> >          5.,   5.,   5.,   4.,   4.,   4.,   4.,   4.,   3.,   3.,   3.,
> >          3.,   3.,   3.,   3.,   3.,   3.,   2.,   2.,   2.,   2.,   2.,
> >          2.,   2.,   2.,   2.,   2.,   2.,   2.,   2.,   1.,   1.,   1.,
> >          1.]
> >
> > There are many classes with just one or two examples in the validation set. As can be seen from the statistics in https://github.com/visipedia/inat_comp/tree/master/2018 things are similarly bad for the iNaturalist dataset. It is hard to believe that overfitting cannot occur with such validation datasets.
> >
> > I am afraid I am still concerned about the scientific novelty of this work as well:
> >
> > Authors say:
> > ""
> > (1) While multiplicative and additive logit adjustments are proposed by recent works, this is the first work (to the best of our knowledge) to demonstrate that combining both can lead to substantial benefits.
> > (2) Unlike earlier works, we demonstrate that small classes can provably benefit from stronger data augmentation, we propose “personalized augmentation”,
> > ""
> > However:
> > (1) Citation [36] already does this. See Eq. (1) in [36] and comments surrounding it. Eq (11) in [50] is also very close to Eq (4.1) in this paper.
> > (2) Using stronger data augmentation for tail classes is a common practice in use to improve the performance on long-tailed datasets. In fact, this practice even predates modern deep learning techniques. See `Nitesh V Chawla, Kevin W Bowyer, Lawrence O Hall, and W Philip Kegelmeyer. Smote: synthetic minority over-sampling technique. Journal of artificial intelligence research, 16:321–357, 2002.` and the large number of papers that cite it.
> >
> > Given the above, I am afraid I am keeping my original rating.

---

> > > ### Comment · Area_Chair_7qxK · 2021-08-26
> > > **Regarding the tails classes in the validation sample**
> > >
> > > Dear Authors,
> > >
> > > Thanks for the detailed responses to the reviewers.
> > >
> > > I am particularly curious about your thoughts on Reviewer mSPX's concern that the validation sample for some the datasets you use may contain classes that are very small. Did you find this problematic in your experiments? If not, do you have strong intuition for why you didn't observe severe overfitting w.r.t. the tail classes in the validation set?
> > >
> > > I understand that the number of hyper-parameters is small relative to the number of parameters in the model and I see that you do have a discussion in the paper on why you are less prone to over-fitting to the validation sample. However, given that you have multiple tunable parameters for each class, don't you risk overfitting on classes that have very few examples in the validation set (when for some of the tail classes, the number of per-class tunable parameters > number of examples)?
> > >
> > > If I understand correctly, in your paper, you mention that you use 20% of the dataset for validation. Just confirming that this means that the validation sample is also as imbalanced as the training set, or could it be that the validation sample is slightly more balanced to facilitate better estimation of the hyper-parameters?
> > >
> > > Regards,
> > >
> > > AC

---

> > > > ### Author Response · Authors · 2021-08-31
> > > > **Tail classes in the validation sample**
> > > >
> > > > Dear Area Chair,
> > > >
> > > > Thank you for reaching out and highlighting Reviewer mSPX's concern on tail classes. We already responded to Reviewer mSPX (specifically Comment 2/4). However; we also wanted to post a short response here.
> > > >
> > > > **Re: Validation sample for some of the datasets you use may contain classes that are very small. Did you find this problematic in your experiments? If not, do you have strong intuition for why you didn't observe severe overfitting w.r.t. the tail classes in the validation set?**
> > > >
> > > > Comment 2/4 to Reviewer mSPX provides Tables A-D that demonstrate that validation overfitting on the tail classes is rather negligible on CIFAR-10/100-LT datasets. We believe the precautions listed on the same comment help mitigate overfitting. We use
> > > >
> > > > - Class clustering
> > > > - Imbalanced validation
> > > > - Standard data augmentation on validation
> > > >
> > > > **Optimization challenge:** We would also like to remark that overfitting to the validation data implicitly takes it granted that the bilevel problem can be optimized well and it has solutions that indeed overfits. For instance, if the optimization gets stuck in a poor minima this could prevent overfitting even if there are more hyperparameters than validation samples. As far as we are aware, the literature lacks a solid understanding of the optimization landscape of bilevel problems with deep networks. This is in contrast to single-level problems where it is known that sufficiently large deep networks can fit the data (both empirically and theoretically).
> > > >
> > > > **Re: However, given that you have multiple tunable parameters for each class, don't you risk overfitting on classes that have very few examples in the validation set.**
> > > >
> > > > We agree with this concern as well. For tail classes, our main precaution is "class clustering" which shares the same hyper-parameters across multiple classes with similar frequencies. Quoting Lines 259-264 of the manuscript:
> > > >
> > > > *"For CIFAR-100, because the size of validation data in minority classes can be too small (e.g., only one example in a tail class), as visualized in Figure 3(a), we gather classes of similar frequencies into clusters of size 10 to share the same hyper-parameters (i.e., the values and updates of $(l_y, \Delta_y)$’s). Similarly, for ImageNet-LT, we gather classes into clusters of size 10, and for iNaturalist, we gather classes into clusters of size 40."*
> > > >
> > > > **Re: Just confirming that this means that the validation sample is also as imbalanced as the training set, or could it be that the validation sample is slightly more balanced to facilitate better estimation of the hyper-parameters?**
> > > >
> > > > The validation sample is indeed imbalanced just like training sample. This is an intentional choice to better estimate the hyper-parameters of the head (non-tail) classes. However, we emphasize that the validation loss is still balanced because we weight the total validation loss of each class inversely with the class frequency.
> > > >
> > > > The intuition behind imbalanced validation is to maximize the oversampling ratio $\frac{\text{num validation samples}}{\text{num hyper-parameters}}$ over each class. Every single example would help towards this. Oversampling ratio is larger for head classes so this would primarily benefit the head hyperparameters. However, by better estimating head, it can also help tail classes by reducing the number of hyperparameters prone to overfitting or by improving the optimization landscape.
> > > >
> > > > We would be happy to address further questions.
> > > >
> > > > Authors

---

> > > > > ### Comment · Area_Chair_7qxK · 2021-08-31
> > > > > **Follow-up question**
> > > > >
> > > > > Dear Authors,
> > > > >
> > > > > I appreciate the detailed reply to our questions. I will let Reviewer mSPX weigh in on your response.
> > > > >
> > > > > If you don't mind, just for my own understanding, I have a couple of follow-up questions. I do know that the discussion period will come to an end very soon (Sep 2nd), and so would totally understand if you are unable to send in a response before the deadline.
> > > > >
> > > > > It looks like the success of you proposed procedure would depend on the size of the validation set (despite the precautions taken, a very small validation sample could be problematic as this would mean that there are hardly any examples seen from the tail classes). Currently, you use 20% of the original training data for validation, and the remaining for training.
> > > > >
> > > > > (1) Do the baselines LDAM and LA loss (with \tau=1) also need a validation sample (for the latter, if \tau is set to a default value of 1, are there other hyper-parameters that need to be tuned using a held-out sample)?
> > > > >
> > > > > (2) If not, to be fair to these methods, should we be running these baselines on the combined train + validation sample? I'm not asking for additional experiments, but trying to understand what the right comparisons are for the proposed approach. In particular, a practitioner might be curious to know if they are better of applying a prior method with default hyper-parameters on the original training dataset, or splitting the data into train and validation samples and applying your method to optimize for the best loss hyper-parameters.
> > > > >
> > > > > Again, I understand if you are unable to get to these questions on time.
> > > > >
> > > > > Thanks,
> > > > >
> > > > > AC

---

> > > > > > ### Author Response · Authors · 2021-08-31
> > > > > > **Re: Follow-up question**
> > > > > >
> > > > > > Dear AC,
> > > > > >
> > > > > > Thank you for giving us a chance to clarify.
> > > > > >
> > > > > > **Short answer:** The LDAM and LA losses in Table 1 are trained on the original training dataset. This original training dataset is equal to the union of the training and validation splits.
> > > > > >
> > > > > > **Longer answer:** We agree that if we don't use the original dataset it is an unfair comparison as they would have access to less examples. This is also why our approach also needs the additional retraining phase on the original dataset (in Figure 3b).
> > > > > >
> > > > > > Your question also relates to our discussion with Reviewer mSPX (see Comment 4/4). From our reading, the related works [8,36,50,63] all tune their hyper-parameter choices on the test data. Arguably, this is the first work compared to these references that actually does the train-validation-test process in the proper fashion. However, since they have very few hyper-parameters (e.g. LA loss only has $\tau$), we believe tuning on test is mostly acceptable and won't significantly affect their results. Using validation-based tuning was more important for us due to our larger hyper-parameter space.
> > > > > >
> > > > > > Thank you,
> > > > > >
> > > > > > Authors

---

> > > > > > > ### Comment · Area_Chair_7qxK · 2021-08-31
> > > > > > > **Re: Follow-up question**
> > > > > > >
> > > > > > > Appreciate the clarification. This is helpful!
> > > > > > >
> > > > > > > -AC

---

> > > ### Author Response · Authors · 2021-08-30
> > > **Comment 4/4: Overall scientific novelty**
> > >
> > > **(4) Re: Concerns on scientific novelty**
> > >
> > > We genuinely care about the reviewer’s concern and would like to address this issue better. At a high-level we claim two (interleaving) contributions. (1) The ability to design/tune the training loss function to optimize a validation objective. (2) Applications to imbalanced data and SOTA results for benchmark datasets on a problem that has attracted significant attention the past few years [8,28,36,41,50,63]. The reviewer raised concerns on (2) which we addressed above. However, we believe delivering (1) (and its benefits on (2)) is also a key contribution of the paper and would like to draw attention to it.
> > >
> > > **[Getting differentiable train-validation procedure to work]:** We believe that every machine-learning practitioner will agree that getting the overall approach and ideas to work is a highly non-trivial effort. In particular, it involves making a number of theoretically proper algorithmic choices. For instance, we use an appropriate warm-up phase (e.g. Fig 3(b,c,d) are initially flat lines until epoch 120) following the discussion on implicit differentiation (Lines 207-219). Without this warm-up phase, the algorithm is unstable and produces poor results. Also, the retraining phase is critical to achieve competitive guarantees compared to [8,36,50,63],  which tune on test data(!). Secondly, we are not advocating naively fitting hyperparameters. As shown in Fig 3(a) and explained earlier, we cluster the classes to reduce the dimensionality of the search space. Here, each color corresponds to a cluster of class IDs which uses the same loss function choice. Without clustering, we observed that our approach was not as robust and may not give competitive results. We also derive Theorem 1 that supports why hyperparameter dimensionality reduction helps and why validation is critical to mitigate overfitting. Note that this also relates to the reviewer’s question on overfitting to validation. In fact, we believe that the reviewer’s concerns on validation-based optimization and overfitting implicitly supports the scientific novelty of our work. We (perhaps counterintuitively) demonstrate that validation-based optimization actually can work and lead to competitive results, and it actually doesn’t overfit nearly as much as training (CIFAR10-LT, CIFAR100-LT train/validation/test tables above).
> > >
> > > **[Properness of the train-validation procedure]:** We again note that the related works [8,36,50,63] all tune their hyperparameter choices on the *test data* ([8] tunes and reports on validation so validation=test, [63] discusses validation only in Table 9 of their supp). Arguably, this is the first work compared to these references that actually does the train-validation-test process in the proper fashion. Here, we don’t intend to say that results of the other papers are not valid because the “amount of overfitting” is small if one has few hyperparameters. However, we would like to draw attention to the scientific correctness of our validation/test evaluation methodology (besides its ability to efficiently optimize over a much larger hyperparameter space).
> > >
> > > **[Learning loss function from scratch and insights]:** We believe that the ability to learn a good loss function from scratch is interesting. It is particularly surprising with differentiable optimization which is more prone to instability (again also see Reviewer FVKf). Our approach manages to do this in an intuitive and robust fashion. Consider subfigures (b),(c),(d) in Figure 3. We learn the loss from scratch (no priors on $l,\Delta$) and the optimization process gracefully evolves to choices that assign larger $\Delta$&$l$ for larger classes (class IDs from [0,19]) compared to smaller classes (class IDs from [80,99]). This is exactly what one would hope to see (following the intuition of smaller classes being penalized more). At a high-level, the ability of SGD to find promising loss function designs (rather than getting stuck at poor solutions) motivates further research into better algorithms and different application scenarios.
> > >
> > > We thank the reviewer again for their thorough review and would be happy to respond to further concerns.

---

> > > ### Author Response · Authors · 2021-08-30
> > > **Comment 3/4: Comparisons to [36,50] and novelty of PDA**
> > >
> > > **(3a) Re: In [36] and comments surrounding it. Eq (11) in [50] is also very close to Eq (4.1) in this paper**
> > >
> > > Comparison to [36]: We have acknowledged that the loss function jointly combining $w,l,\Delta$ is first introduced in [36]. As a matter of fact, in Line 153, we explicitly state regarding the loss in Eq. (4.1): “This loss is the same as VS-loss of [36], which also contains $w,l, \Delta \in R^K$”. Also see Lines 89 and 101 which further highlights this. Also, please note that our “proposed loss function” that eventually enters our algorithm is given not in Eq (4.1) but in Eq. (4.2), that is a slight refinement of (4.1) where we add $\sigma$ (sigmoid applied to $\Delta$’s) and ${\cal{A}}$ (data augmentation) terms. Specifically for the sigmoid applied to multiplicative $\Delta$ terms, although minor adjustment, we have found empirically that ensuring nonnegativity of $\Delta$’s is crucial for our algorithm’s convergence.
> > >
> > > Also we noticed that there is a very recent version of [36] posted just a few days ago. It is not clear which version the reviewer is referring to so we briefly discuss both versions [36v1] and [36v2]. [36v1] is the version at the time of our submission. In this version, [36] still proposes (4.1) (with slight variation, see our Line 154). However, the focus throughout the paper is on showcasing the benefits of multiplicative adjustments $\Delta$. Specifically, Proposition 1 in [36v1] establishes an equivalence between VS-loss and cost sensitive(CS)-SVM for linear models. Note that CS-SVM ignores the additive term $l_y$ and only cares about $\Delta_y$. In summary, while we demonstrate the  the synergistic benefit of all three hyper-parameters $(w_y,l_y,\Delta_y)$, [36v1] focused on the benefit of $\Delta_y$ over $l_y$. Here, we note that while we optimize $l_y,\Delta_y$, we fix $w_y=1/\pi_y$ for both train and validation phases. This choice is particularly useful for ensuring unbiasedness of the imbalanced validation as discussed earlier. Additionally, their only real-data class-imbalanced experiment is on MNIST using random features rather than neural nets. Instead, our experiments are all on deep-nets and on state-of-the-art (SOTA) datasets.
> > >
> > > We now turn to [36v2], which was posted just a few days ago. Despite that, we will still discuss our differences with them (both here and in the camera-ready). In short, we believe that there are unique contributions in both works that are in some sense complementary to each other: their focus is on the theoretical/analysis component, while our focus is on the algorithmic front and on obtaining SOTA results on large-scale datasets. Concretely, while their new version also argues in favor of the *joint* benefits of $l_y$ and $\Delta_y$, there are important differences. First, for tuning they simply combine the choices described in CDT [63] and LA [50] papers. Instead, we introduce the aforementioned refinements in Eq. (4.2), a sophisticated tuning strategy,  data-splitting strategies for hyperparameter grouping and algorithm initialization strategies using LA-loss or wCE. Note also that all these algorithmic refinements are applicable not only on VS-loss, but also on CDT- and LA-losses individually. Additionally, unlike us, they don’t report any result on large-scale SOTA datasets (see Imagenet, iNaturalist). Overall, even comparing to contemporaneous work,  we are still the first (to the best of our knowledge) to show the joint benefit on these datasets and the first to experiment and report auto-tuning results of any logit-adjusted loss. Finally, as mentioned in their experimental section, their reported values in Tables 1 and 2 are results on *validation set*, not on the test set. In fact, this non-realistic strategy (aka tuning on validation and testing on validation => validation=test) is also followed in all preceding works on logit-adjusted losses [8,36,50,63]. Instead all our results follow the scientifically formal strategy of tuning on validation, testing on test.  We will make sure to acknowledge [36v2] in our revision.
> > >
> > > Comparison to [50]: The reviewer states **“Eq (11) in [50] is also very close to Eq (4.1) in this paper.”**. This is not correct. Observe that this is simply additive adjustments where $Delta_{yy’}=l_{y’}-l_y$. Perhaps what confused the reviewer is that their $\Delta$ corresponds to our $l$ and also they state their Equation (11) as a multiplication of exponents.
> > >
> > > **(3b) Re: Using stronger data augmentation for tail classes**
> > > Thank you for pointing us to Smote [9]. We were aware of this work in the context of over-sampling methods, but we were unaware of the augmentation aspect. They are indeed proposing a variation of mixup (much earlier than the mixup paper [Zhang et al. ICLR’18]). We will highlight and acknowledge [9] and other works that bring up this idea. That said, we believe this doesn’t take away from our current set of contributions, because PDA is only a part of our design space Eq. (4.2), rather than its main component. We also believe our results regarding PDA are novel because they all relate PDA to VS-loss or evaluate PDA on top of the VS-loss. Specifically, first, in Lemma 2 we establish an equivalence between PDA and VS-loss. Given that VS/CDT/LA losses are new (we also checked their citations), this should be the first result that formally relates the benefits of PDA to logit adjustment loss. Second, the PDA experiments (only) appear at Table 2. The goal of the second line $\alpha\gets \text{PDA}$ is verifying our Lemma 2. Because while for simple settings (like Lemma 2), it is intuitive that PDA helps (see Fig. 2), it is unclear if it still has benefit (compared to Line 1 using non-personalized DA) if the augmentation policy is optimized over a rich search space (such as the AutoAugment, MADAO papers). Line 2 verifies that intuitions from Lemma 2 carry over to these more sophisticated settings. Finally, Line 3 $\alpha\gets \text{PDA}$&$\Delta$&$l$ evaluates the combined benefit. This is also discussed in response to Reviewer WkxY (final comment). We found that PDA on top of $\Delta$&$l$ indeed helps (compare Line 3 of Table 2 to the second to last line of Table 1) which we believe is a novel contribution. However, most of the benefit is thanks to $\Delta$&$l$.

---

> > > ### Author Response · Authors · 2021-08-30
> > > **Comment 2/4: Why our measures can mitigate overfitting to validation and numerical results on the tail of CIFAR-10&100-LT**
> > >
> > > **(2) Re: There are many classes with just one or two examples in the validation set. As can be seen from the statistics in https://github.com/visipedia/inat_comp/tree/master/2018 things are similarly bad for the iNaturalist dataset. It is hard to believe that overfitting cannot occur with such validation datasets.**
> > >
> > > We agree with the reviewer that the long-tail portion of the data is of particular concern. We first adapt the high-level analysis above to the long-tail setting and discuss our precautions. We then provide numerical results on long-tail overfitting in CIFAR.
> > >
> > > **$\bullet$ High-level analysis for balanced error and our precautions:** Above is the argument for assessing the overfitting for standard error. However, let us briefly discuss what happens for the balanced error. Recall that in the paper, we specifically use imbalanced validation data. Our intent was in line with the reviewer’s concern on mitigating overfitting (at least in the larger classes). However, for balanced analysis, let us discard the validation data from larger classes and assume that each of the $K$ classes has the same size as the smallest class with $n_{\text{small}}$ samples. Then, applying the above argument, we need $n_{val}=Kn_{\text{small}}\gtrsim O(h)$. In our case, $h$ is typically equal to $2K$ ($K$ for $\Delta$ and $K$ for $l$) so if $n_{\text{small}}$ is 1, we would be in trouble. However, we do the following actions to mitigate overfitting:
> > >
> > > $\circ$ **Precaution (1) Class clustering:** We cluster the classes by frequency and use the same $\Delta_y$ and $l_y$ on the same cluster (color coding in Figure 3(a)). Suppose cluster size is $C$. With this, dimensionality of $h$ becomes $2K/C$ rather than $2K$. This means that the oversampling ratio $n_{val}/h$ is proportional to $C$ and can be kept high by choosing a larger $C$. For instance, for CIFAR-100, $n_{\text{small}}=1$ however we select $C=10$. This way, the oversampling ratio is nearly the same as CIFAR-10 where we select $C=1$ and $n_{\text{small}}=10$. Regarding this, we quote Lines 259-264 of the manuscript:
> > >
> > > *For CIFAR-100, because the size of validation data in minority classes can be too small (e.g., only one example in a tail class), as visualized in Figure 3(a), we gather classes of similar frequencies into clusters of size 10 to share the same hyper-parameters (i.e., the values and updates of (ly, ∆y)’s). Similarly, for ImageNet-LT, we gather classes into clusters of size 10, and for iNaturalist, we gather classes into clusters of size 40.*
> > >
> > > Aligned with the reviewer’s intuition, for iNaturalist we use larger clusters of size 40.
> > >
> > >
> > > $\circ$ **Precaution (2) Imbalanced validation:** Instead of discarding the validation samples of large classes (as above), we use imbalanced weighted validation data. This increases the oversampling ratio over large classes. However, it can also help small classes by reducing the number of hyperparameters prone to overfitting or prone to unstable optimization.
> > >
> > > $\circ$ **Precaution (3)** We also apply data augmentation on the validation data, which intuitively further mitigates overfitting risk. We remark that over validation, we used standard augmentation and it is not personalized.
> > >
> > > **$\bullet$ Numerical results on CIFAR**
> > >
> > > The reviewer’s new feedback helped us understand the long-tail concern better. To this end, we would like to report the train/validation/test results where we slice the CIFAR-10-LT and CIFAR-100-LT datasets by the class frequencies. The results are obtained by averaging three independent runs. The results use the “Algo. 1$\gets \Delta$&$l$” strategy in Table 1, i.e. we are optimizing $\Delta$&$l$ parameters. Since we report the train/validation/test errors, these evaluations are for the model obtained after the Search Phase is completed (see Figure 1(b)).
> > >
> > > The two tables below are for CIFAR-100-LT. Since we use a cluster size of 10 for CIFAR-100-LT, each slice corresponds to the class-averaged error in one of the 10 clusters.
> > >
> > > **Table A: CIFAR100-LT, Balanced errors in 10 slices**
> > >
> > > |                  | 0-9              | 10-19            | 20-29            | 30-39            | 40-49            | 50-59            | 60-69            | 70-79            | 80-89            | 90-99            |
> > > |------------------|------------------|------------------|------------------|------------------|------------------|------------------|------------------|------------------|------------------|------------------|
> > > | Training error      | 3.91$\pm$0.24  | 3.04$\pm$0.16  | 0.84$\pm$0.17  | 0.73$\pm$0.20  | 0.29$\pm$0.16  | 0.20$\pm$0.16  | 0.00$\pm$0.00  | 0.00$\pm$0.00  | 0.00$\pm$0.00  | 0.00$\pm$0.00  |
> > > | Validation error | 43.56$\pm$0.90 | 51.98$\pm$0.95 | 46.55$\pm$1.28 | 55.79$\pm$2.48 | 57.01$\pm$2.49 | 59.42$\pm$0.91 | 67.50$\pm$0.68 | 64.72$\pm$3.18 | 76.67$\pm$6.71 | 88.33$\pm$5.93 |
> > > | Test error       | 40.87$\pm$0.38 | 51.97$\pm$1.09 | 46.80$\pm$0.59 | 56.83$\pm$0.56 | 57.10$\pm$0.71 | 58.67$\pm$0.90 | 68.93$\pm$0.91 | 75.93$\pm$0.87 | 74.90$\pm$2.55 | 80.13$\pm$0.68 |
> > >
> > > The top row displays the class ID’s that belong to a cluster/slice. Observe that training error is very small and it is consistently zero on the tail classes 60-99.  In contrast, the validation error (averaged over three runs and 10 classes within the cluster) is consistently nonzero and close to the test error. Observe that the standard deviations of the validation errors are much larger than that of the test especially on the tail classes. This is due to the small sample size. On the 90-99 slice, there are around 10-20 validation examples per run.
> > >
> > > However, if we gather 10 slices into 2 slices of 0-49 and 50-99, we have a more clear view.
> > >
> > > **Table B: CIFAR100-LT, Head & Tail balanced error**
> > >
> > > |                  | 0-49  | 50-99 | All   |
> > > |------------------|-------|-------|-------|
> > > | Train error      | 1.76  | 0.04  | 0.90  |
> > > | Validation error | 50.98 | 71.33 | 61.15 |
> > > | Test error       | 50.71 | 71.71 | 61.21 |
> > >
> > > This table shows that validation error is very close to the test error and provides good evidence that overfitting is negligible in the tail classes as well as the whole validation set. Also note that head classes have around 2% training error whereas tail is around 0%.
> > >
> > > **Inaccuracy caught in our initial response:** The reviewer will notice that the last column of Table B (which reports overall balanced error) is not consistent with what we reported in our initial response. Specifically, we reported training error to be 2.46% and validation error to be 48.52%. This happened because we mistakenly reported standard (imbalanced) errors on the augmented train and validation sets. CIFAR-10-LT also had the same issue. We have corrected these issues now, and report everything on balanced error with actual samples. Our apologies for this inaccuracy.
> > >
> > > Below we display the same tables for CIFAR-10-LT. Each slice is exactly one of the classes.
> > >
> > > **Table C: CIFAR-10-LT, Balanced errors per class**
> > >
> > > |                  | 0     | 1    | 2     | 3     | 4     | 5     | 6     | 7     | 8     | 9     |
> > > |------------------|-------|------|-------|-------|-------|-------|-------|-------|-------|-------|
> > > | Train error      | 1.29$\pm$0.12 | 1.26$\pm$0.39 | 0.16$\pm$0.05  | 0.46$\pm$0.09  | 0.26$\pm$0.14  | 0.00$\pm$0.00  | 0.00$\pm$0.00  | 0.00$\pm$0.00  | 0.00$\pm$0.00  | 0.00$\pm$0.00  |
> > > | Validation error | 8.40$\pm$0.43 | 6.01$\pm$1.03 | 18.57$\pm$1.63 | 30.39$\pm$0.89 | 19.64$\pm$2.57 | 34.20$\pm$2.15 | 24.64$\pm$5.26 | 21.43$\pm$4.45 | 29.17$\pm$6.13 | 30.00$\pm$4.71 |
> > > | Test error       | 8.27$\pm$0.67 | 4.50$\pm$0.49 | 20.27$\pm$0.94 | 33.07$\pm$1.44 | 24.07$\pm$0.48 | 33.27$\pm$2.98 | 23.97$\pm$0.14 | 28.87$\pm$1.56 | 30.80$\pm$0.29 | 27.80$\pm$2.19 |
> > >
> > > **Table D: Cifar10 Head & tail balanced error**
> > >
> > > |                  | 0-4   | 5-9   | All   |
> > > |------------------|-------|-------|-------|
> > > | Train error      | 0.69  | 0.00  | 0.34  |
> > > | Validation error | 16.60 | 27.89 | 22.24 |
> > > | Test error       | 18.03 | 28.94 | 23.49 |
> > >
> > > These tables similarly support that overfitting is negligible. However, we observe that CIFAR-10-LT seems to have a larger “test error - validation error” gap (i.e. amount of overfitting) compared to CIFAR-100-LT. While we don’t have a strong intuition, this might be because CIFAR-100-LT is more difficult to optimize due to containing a larger number of classes. For instance, the overall balanced training error on CIFAR-10-LT (0.34%) is also smaller than that of CIFAR100-LT (0.9%). Finally, we re-emphasize that due to class clustering in CIFAR-100-LT, the hyperparameter oversampling ratio $n_{\text{val}}/h$ is essentially the same for both experiments.
> > >
> > > We hope this response clarifies the concerns on overfitting. However, if the reviewer believes they can benefit from more experiments (regarding the overfitting concern), we are more than happy to run them (assuming we can finish them in time for the review period) and report the results.

---

> > > ### Author Response · Authors · 2021-08-30
> > > **Comment 1/4: Thanks and Response**
> > >
> > > Thank you again for your feedback.  This provided us a better understanding of your concerns on overfitting to tail classes. We respond to your comments in order. For your reference, we are posting four comments:
> > >
> > >
> > > - **This comment:** Where we clarify how validation phase helps (by being agnostic to the number of model parameters)
> > > - **Comment 2:** Why our measures can mitigate overfitting to validation and numerical results indicating negligible long-tail overfitting in CIFAR
> > > - **Comment 3:** Comparisons to [36,50] and novelty of PDA
> > > - **Comment 4:** Overall scientific novelty
> > >
> > > We attempted to post a thorough response. Thus, we apologize for the inconvenience if you find parts of the discussion below overlap with some of our earlier comments.
> > >
> > > **(1) Re: I believe any machine learning researcher/practitioner would agree that overfitting to the validation set is a real issue in practice.**
> > >
> > > We believe for the most part we are on the same page with the reviewer. We agree that one can potentially overfit to the validation in practice. We also agree that overfitting to the long tail is an additional concern. In an earlier response, we mentioned that “... it is difficult to overfit to the validation set when the number of hyper-parameters is much less than the validation size”.
> > >
> > > In the following discussion, we will be more specific about what overfitting means. It typically means almost perfectly fitting to the training data. However, for our discussion, let us quantify it via “excess risk” definition
> > >
> > > $$\text{amount of overfitting} = \text{empirical error}- \text{test error}$$
> > >
> > > Empirical error corresponds to either training or validation errors. With this definition, the ``amount of overfitting’’ is a soft quantity and, intuitively, some amount of overfitting can always occur as long as we are dealing with finite (training or validation) samples.
> > >
> > > We believe the disagreement might be arising from the reviewer’s following concern (quoting them)
> > >
> > > “With millions of parameters in the actual model and a handful of hyper-parameters, it is easy to overfit to the validation set as well, especially when the validation set is small.”
> > >
> > > As also detailed in our earlier response, we advocate that
> > >
> > > *Claim 1:*  In the setting of
> > > ***
> > > millions of parameters >> size of the validation dataset > handful of hyper-parameters
> > > ***
> > > the amount of overfitting during the validation phase is small; thus the validation error is a meaningful indicator of the test error.
> > >
> > > **High-level analysis:** The key reason behind Claim 1 is that *the amount of overfitting in the validation phase has minimal dependence on the millions of training parameters; instead it is governed by only a handful of hyper-parameters*. Theorem 1 in the Appendix formalizes this argument for continuous-valued hyperparameters. But again,  we emphasize that this is perhaps easiest argued by considering a discretization of  the hyperparameter space. Suppose each hyperparameter is searched on a grid $[-C/2,C/2]$ with grid spacing $\delta$ and we evaluate the validation error of each trained model. In total there are $(C/\delta)^h$ configurations where $h$ is the number of hyperparameters. Independent of how many training parameters there are, since validation examples are independent, we can apply Hoeffding + Union Bounds to conclude with the following:
> > >
> > > *Claim 2:* If the total validation size $n_{\text{val}}$ obeys $n_{\text{val}}\geq O(\epsilon^{-2}h\log(C/\delta))$, then
> > >
> > > $$\text{amount of validation overfitting}:=\text{validation error}- \text{test error} \leq \epsilon$$
> > >
> > > with probability $1-(C/\delta)^{-h}$ over the validation data. Thus, $n_{\text{val}}$ does not need to depend on the # of training parameters and it essentially grows as $O(h)$. The upper bound on the amount of overfitting grows as $(n_{\text{val}}/h)^{-0.5}$, where $n_{\text{val}}/h$ is the “oversampling ratio”.

---

### Official Review · Reviewer_FVKf · 2021-07-24

**Rating:** 7
**Confidence:** 4

**Summary:**

The paper proposes and empirically validates a way to use - and tune - high-dimensional hyperarameters affecting training loss, which are used for fairness-seeking test objectives.

**Ethical Concerns:**

I do not see ethical issues with the paper.

**Limitations And Societal Impact:**

The authors have adequately addressed the potential negative societal impacts of their work.

The authors present reasonable limitations.

**Main Review:**

To the best of my knowledge, the proposed method is novel. My experience is primarily in high-dimensional bilevel optimization, and the related work looks adequately cited there.  I am less able to comment on related work long-tailed learning or group-sensitive and fair learning.

The paper's bilevel optimization framework seems technically sound.  The gradient estimates seem reasonable, and I’m particularly impressed by the authors results with the final re-training phase with fixed optimal hyperparameters.

A contribution the authors did not emphasize, but I think is present is the following. It is common to neglect this training phase, and many models fail to re-train to reasonable performance with the final hyperparameters due to hysteric optimization effects.  The hysteric effects are particularly strong when tuning the loss.  However, the authors circumvent this with their loss parameterization. It would be interesting to try to generalize these techniques to more general setups than fairness, or understand this phenomena more abstractly. For example, by building on Lemma 2.

I think the paper suffers from various issues with clarity.

The text from 207-219 is unclear and the notation seems inconsistent.  Is \theta^* a function or a point?  Where are derivatives evaluated at?  What do you mean by “Also, observe …. Computed by taking the gradient”?

You discuss the importance of a train-val split in 220-233, but I do not see a discussion of how you tuned the split for this problem?  You also discuss the effects of overparameterization, but its not clear to me what kind of sizes of alpha, theta, train size, val size, etc. are actually used in practice.

I find it difficult to read objects like table 1, table 4 due to the dense notations.  I would appreciate a notation table summarizing it.

The paragraph from line 39-58 is long and dense.  Perhaps the ideas should be modularized.  The results and discussion from 273-299 is similar.

Many components of line 252-267 could probably be moved to the appendix.

I believe the results are reasonably significant because they propose a useful way to introduce high-dimensional hyperparameters controlling the loss that still work even after retraining.  The results seem useful for fairness seeking objectives too. Practitioners are likely to use the method, because it is simple to understand, does not look difficult to implement, does not introduce excessive computational burden, and seems to work reasonably well. The paper also claims to advance the state-of-the-art, which seems supported by the experiments to the best of my knowledge.

My score is 7, because I think the paper makes good contributions to learning losses with fairness seeking objectives. My confidence is a 3, because I am not an expert on fairness seeking objectives. The score would be improved to an 8 if the authors largely improved the clarity.


**Time Spent Reviewing:**

7

---

> ### Author Response · Authors · 2021-08-10
> **Response to Reviewer FVKf**
>
> We greatly appreciate your thorough review and positive comments. Also thanks for the organizational feedback. We address the comments point-by-point.
>
> **(1) Re: “hysteric effects & generalizations”.** Thank you for pointing this out. Indeed the optimization process is sensitive to the loss function hyperparameters and requires care, especially with fewer validation samples. This is also why we use a warm-up phase, class clustering, and apply a sigmoid function in Eq (4.2). We also absolutely agree that the loss function design for general objectives (beyond our fairness/imbalance motivation) is a promising future research direction.
>
> **(2) Re: “The text from 207-219 is unclear”.** Indeed, we should have been more clear here and we will revise accordingly. As displayed in Fig. 1 and Algo. 1, $\theta$ is the parameters of the model $f_{\theta}$. $\theta^\star$ is a function of $\alpha$. In essence, it is the final weights obtained by training with hyperparameters $\alpha$ (i.e. solution of the lower-level problem). For the “also observe that…” sentence, we meant to say that the derivative of the loss with respect to the weights $\theta$ is the usual gradient, which is a standard computation, and the main challenge lies with the derivative of $\theta^\star(\alpha)$ with respect to $\alpha$.
>
> **(3) Re: “Discussion on tuning the split, practical sizes of alpha, theta, train size, val size”.** For the train-validation split (see Line 250), we used 80%-20% ratio because for CIFAR100-LT, 20% is the minimum proportion to ensure that each class has at least 1 example. In general, we did not really attempt to tune the ratio of the training-validation split, and we also believe that other reasonable splits (e.g. 70-30 or 60-40) will lead to comparable results. For the effect of overparameterization, the hyper-parameter $\alpha$ has 3 parameters for each class cluster $(\Delta_i, l_i, w_i)$ and if the data augmentation is applied, 42 additional parameters are optimized. As the weights $\theta$, in CIFAR experiments (see Line 252), we used a ResNet-32 model with 0.47M parameters. We discuss the formation of the dataset in Line 241. We exponentially reduce the sample size per class to obtain long-tailed data. For CIFAR100-LT, this leads to 8716 training samples and 2183 validation samples. We plan to use the additional 1-page space (provided for accepted papers) to provide all the missing details on these.
>
> **(4) Re: “Notation table.”** Good suggestion. First, we will better clarify the notation $\alpha \gets XXX$ and better highlight the validation loss in Line 307 of Sec. 6, which involves the $\lambda$ term. We will also add a table briefly defining the multiplicative term “$\Delta$”, additive term “$l$”, and “PDA”.
>
> **(5) Re: “39-58 is dense and similar to 273-299”.** We agree that 39-58 contains redundancies. We will remove/shorten the Lines 39-45 and Lines 51-55 to improve the flow. All citations will be preserved.
>
> **(6) Re: “Move 252-267 to appendix.”** We will move the standard or repeated details to the appendix and use the extra space for additional experimental results or more relevant discussion (such as the reviewer’s comment above on “tuning the split”).

---

> > ### Comment · Reviewer_FVKf · 2021-08-23
> > **Thanks for your response.  I have read the other reviews, and will maintain my score of 7 and confidence of 4**
> >
> > Thank you for the response.  I've read the other reviews, and I'm inclined to keep my score of accepting with a confidence of 4.

---

### Author Response · Authors · 2021-09-01
**Thank you for your time**

Dear Reviewers and Area Chair,

We would just like to thank all of you for the time and effort you spent reviewing our paper, providing many helpful suggestions, following up with the discussion phase and giving us the opportunity to clarify potential concerns. We know that the review process has been very involved this year.

Thanks,

Authors

---

### Decision · Program_Chairs · 2021-09-27

**Decision:**

Accept (Poster)

**Comment:**

The paper proposes a bi-level optimization framework to design a training loss for long-tail and group-fair learning. The reviews attracted a a lot of back-and-forth discussion with the authors, and I appreciate the authors for providing very detailed responses and additional experimental results.

While there were concerns raised about the proposed approach being prone to overfitting to the validation sample, I think the authors have satisfactorily explained how they take precautions to avoid it. I think Reviewer mSPX's concern about there being limited novelty does have some merit, but given that this is one of the first few works to successfully engineer an elegant loss-tuning procedure for long-tail learning and that the experimental results are significant, I would recommend an accept.

Having said this, I strongly urge the authors to use the feedback provided to improve their manuscript, and in particular include the following promised additions to the final version:
- a detailed discussion on the risk of over-fitting and the precautions taken to avoid it (do report the validation errors, perhaps in the appendix)
- comparison to other Bayesian optimization approaches
- comparison to the additional baselines mentioned by Reviewer GFgA

Finally, here are some additional references on prior work on learning loss functions for specialized tasks, which may be of some relevance:

https://arxiv.org/pdf/1905.10108.pdf

https://arxiv.org/pdf/1803.09050.pdf